# PROCESS REWARD MODEL WITH Q-VALUE RANKINGS

**Wendi Li**
Department of Computer Science
Huazhong University of Science and Technology
wendili@hust.edu.cn

**Yixuan Li**
Department of Computer Sciences
University of Wisconsin-Madison
sharonli@cs.wisc.edu

## ABSTRACT

Process Reward Modeling (PRM) is critical for complex reasoning and decision-making tasks where the accuracy of intermediate steps significantly influences the overall outcome. Existing PRM approaches, primarily framed as classification problems, employ cross-entropy loss to independently evaluate each step's correctness. This method can lead to suboptimal reward distribution and does not adequately address the interdependencies among steps. To address these limitations, we introduce the *Process Q-value Model* (**PQM**), a novel framework that redefines PRM in the context of a Markov Decision Process. PQM optimizes Q-value rankings based on a novel comparative loss function, enhancing the model's ability to capture the intricate dynamics among sequential decisions. This approach provides a more granular and theoretically grounded methodology for process rewards. Our extensive empirical evaluations across various sampling policies, language model backbones, and multi-step reasoning benchmarks show that PQM outperforms classification-based PRMs. The effectiveness of the comparative loss function is highlighted in our comprehensive ablation studies, confirming PQM's practical efficacy and theoretical advantage. Our codes can be found at https://github.com/WindyLee0822/Process_Q_Model.

## 1 INTRODUCTION

Process reward modeling (PRM) plays a crucial role in tasks where the quality of intermediate steps is pivotal to achieving the final outcome (Lightman et al., 2024). In complex problem-solving scenarios, such as mathematical reasoning or multi-step decision-making (Shao et al., 2024; Yu et al., 2024; Hao et al., 2024), the accuracy and effectiveness of each intermediate action can significantly influence the overall success. Unlike outcome reward models (ORM) (Cobbe et al., 2021), which focus solely on the final result, PRM provides detailed feedback at each stage of the process. By capturing the value of intermediate steps, PRM allows for a deeper understanding of how each action contributes to the overall goal. This granular approach supports the development of more sophisticated and reliable systems that can navigate complex tasks with greater accuracy.

Existing research typically frames PRM as a classification problem (Wang et al., 2023a; Shao et al., 2024; Lightman et al., 2024; Luo et al., 2024), where each intermediate state is classified as correct or incorrect. Specifically, for a trajectory $\{x, a_1, a_2, \ldots, a_H\}$ where $x, a, H$ represent a question, a reasoning step, and the trajectory horizon, a reasoning state $s_i = (x, a_{1:i-1})$ comprises the instruction $x$ and text pieces previously generated (e.g. reasoning steps in reasoning tasks). Current research uses cross-entropy loss to maximize the probability $p(c_i|s_i)$ for each reasoning state, where $c_i$ is the label indicating whether $s_i$ is correct. While this approach has shown empirical success, it has notable limitations. Classification-based methods treat each state *independently* and do not account for the dependencies and nuances among states within a trajectory. This can lead to suboptimal reward assignments, as these methods often ignore the relative importance of different steps and their influence on the overall process. Furthermore, these approaches lack theoretical grounding on how they approximate the desired reward function.

To address the challenges, we propose a novel framework—Process Q-value Model (**PQM**)—which frames PRM as a $Q$-value ranking problem. This framework allows us to capture the interdependencies among states and provides a more nuanced evaluation of each step's contribution to the overall process. Specifically, our framework is grounded in the Markov Dynamic Process, where each ac-

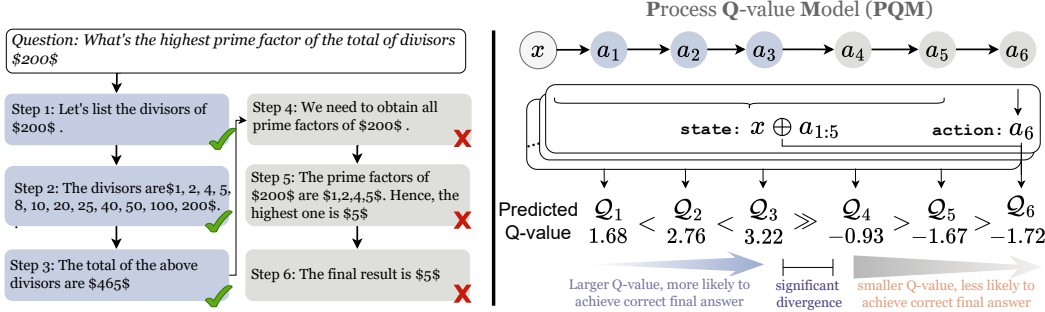

Figure 1: Illustration of our proposed framework **Process Q-value Model** (PQM). The example highlights a solution trajectory with six steps, where the first three steps are correct and the last three steps are incorrect.

tion $a_h$ is a text piece generated based on the current state $s_h = (x, a_{1:h-1})$. The LLM policy $\pi(a_h|x, a_{1:h-1})$ maps the observed state to a distribution over the action space. The process reward model intuitively scores each action $a_h$ based on the instruction $x$ and previous generations $a_{1:h-1}$. In the context of reasoning tasks, we introduce a $Q$-value function for each state-action pair $(s_h, a_h)$ as the probability of success in achieving the correct final answer. Importantly, the $Q$-value function implicitly defines a reward function for intermediate steps. Under this characterization, we formally derive the optimal $Q$-value rankings among reasoning steps, by which we then train PRMs to approximate these rankings with a specialized comparative loss function. According to **Theorem** 3.5, $Q$-values ascend with the continuation of correct steps and descend as wrong steps proceed, while having a prominent gap between correct and wrong steps (see Fig. 1). We further prove that the previous classification-based PRM can be cast as a special case of our theoretical framework under certain conditions.

We conduct comprehensive experiments, revealing the significant advantages of the PQM over prior methods. Following prior research (Wang et al., 2023a; Lightman et al., 2024; Luo et al., 2024), we evaluate PRMs based on their verification ability through best-of-$n$ sampling. The metric assesses the correctness of the most preferred trajectory selected by the PRM from $n$ candidates for each question. Compared to classification-based PRMs, our ranking-based method PQM demonstrates superior accuracy in verification, highlighting its effectiveness in capturing nuanced dependencies among steps. For example, when verifying solutions sampled from the Llama-3-70B-Instruct model, PQM improves the accuracy from 39.8% to 51.4%, a direct 11.6% improvement on the challenging MATH500 benchmark (Hendrycks et al., 2021). These results are consistent across diverse datasets, sampling policies, and LLM backbones, underscoring PQM's effectiveness and generalizability.

To summarize, our main contributions are as follows:

1. We present a new framework for PRM by framing it as a $Q$-value ranking problem, providing a theoretical basis for process reward modeling that captures inter-dependencies among reasoning states. We also show that prior classification-based PRM can be cast as a special case under our framework.

2. We offer a detailed theoretical analysis of PQM and validate its effectiveness through comprehensive experiments on a wide range of sampling policies, LLM backbones, and different test sets.

3. We perform extensive ablation studies on the proposed comparative training objective, and analyze its variations to understand their impact on the model's performance and design.

## 2 PRELIMINARIES

**LLMs for reasoning.** Large language models have demonstrated impressive abilities on challenging reasoning tasks across a wide range of math, science, and coding challenges. Chain of thought (Wei et al., 2022) and related techniques (Wang et al., 2023b; Yao et al., 2023; Besta et al., 2024a;b) have emerged as dominant methods, linking the question and the final answer by a series of intermediate reasoning steps. For a given question $x$ and its corresponding answer $y$, extensive studies (Wei et al., 2022; Chen et al., 2023; Yao et al., 2023; Besta et al., 2024a;b) have shown that prompting LLMs to arrive at solutions via *intermediate steps* $\{a_1, a_2, \dots\}$ can produce more interpretable and accurate results. To generate the final answer, each intermediate step is sampled in an auto-regressive manner:

$a_t \sim \pi_\theta(\cdot|x, a_{1:t-1})$, where $\pi_\theta$ denotes an LLM policy parameterized by $\theta$. The final answer is then generated by $y \sim \pi_\theta(\cdot|x, a_1, a_2, \cdots)$. Note that the final answer can be considered the last reasoning step, so we omit $y$ in our subsequent discussion.

**ORM *vs.* PRM.** Outcome reward model (ORM) and process reward model (PRM) represent two distinct approaches to reward assignment in decision-making tasks, particularly in the context of reinforcement learning and language models. ORMs focus on the final outcome, assigning rewards based *solely on the end state* (Cobbe et al., 2021), which is advantageous when the end goal is clear and well-defined. For example, this approach has been popularly used in LLM alignment frameworks for learning human preferences, where the emphasis is on aligning the model's final output with human judgments (Ouyang et al., 2022; Lee et al., 2023; Rafailov et al., 2024b; Khanov et al., 2024; Im & Li, 2024). However, ORMs often overlook the nuances of the process that lead to the final outcome, potentially ignoring valuable information embedded in the intermediate steps for multi-step reasoning tasks (Uesato et al., 2022).

In contrast, OpenAI's recent work on PRM (Lightman et al., 2024) has shown promise in assigning rewards based on the quality or characteristics of the *intermediate steps*. PRMs are particularly useful in tasks that require complex reasoning or multi-step problem-solving, where the path taken to reach the solution is as important as the solution itself. By rewarding intermediate steps, PRMs can encourage more interpretable and structured problem-solving processes, offering a more granular training signal that captures the intricacies of the decision-making process.

**Process reward modeling with BCE loss.** For a question and a trajectory with several steps, $\tau = (x, a_1, a_2, \ldots, a_H)$, current research on process reward models (Wang et al., 2023a; Shao et al., 2024; Lightman et al., 2024; Luo et al., 2024) typically frames PRMs as a classification problem. This approach aims to maximize the predicted correctness of each reasoning state using a binary cross-entropy (BCE) loss,

$$\mathcal{L}_{\text{BCE}}(\tau) = -\frac{1}{H} \sum_{i=1}^{H} \big(c_i \log p_\theta(c_i|s_i) + (1 - c_i) \log(1 - p_\theta(c_i|s_i))\big), \tag{1}$$

where $c_i$ is the gold classification label of $i$-th step, equal to 1 when $s_i$ is a correct intermediate state otherwise 0. Despite its effectiveness, BCE loss treats each intermediate state independently and does not account for the interdependencies among the reasoning states within a trajectory. By treating each state *in isolation*, BCE loss overlooks the relative contribution each step makes. Moreover, the theoretical support for PRM formulation is also lacking. These limitations motivate our approach of formulating process reward modeling as a $Q$-value ranking problem grounded in the Markov Dynamic Process, where the focus shifts to evaluating the relative quality of different steps in a solution trajectory, thus capturing the interdependencies among steps and providing a more holistic approach to reward assignment.

## 3 PQM: PROCESS REWARD MODEL WITH Q-VALUE RANKINGS

In this section, we introduce our framework **PQM**, which frames process reward modeling as a $Q$-value ranking problem. In what follows, we first define a $Q$-value function for reasoning tasks, which implicitly defines a reward function for each intermediate step (Section 3.2). Then, we derive the desirable $Q$-value rankings among intermediate reasoning steps (Section 3.3), by which we can train PRMs to approximate the intermediate $Q$-values by a comparison-based loss (Section 3.4). Lastly, we demonstrate that classification-based PRMs can be viewed as a special case within our theoretical framework (Section 3.5).

### 3.1 DETERMINISTIC MDP FOR LLMS

**Formulations of MDP.** A standard Markov Dynamic Process can be formulated as $M = (\mathcal{S}, \mathcal{A}, \mathcal{T}, r, \rho, \mathcal{H})$, where $\mathcal{S}$ is the state space, $\mathcal{A}$ is the action space, $\mathcal{T} : \mathcal{S} \times \mathcal{A} \to \Delta(\mathcal{S})$ is the transition kernel, $r$ is the reward function, $\rho$ denotes the initial state distribution, and $H$ is the maximal number of interaction steps. A policy in MDPs, denoted by $\pi : \mathcal{S} \to \Delta(\mathcal{A})$, maps each state to a distribution over actions. The interaction between the environment $M$ and the agent can be described as follows. Initially, the starting state $s_1$ is sampled from the initial distribution $\rho$. At each step $t$, the agent observes the current state $s_t$ and selects an action $a_t$ based on its policy. The

environment then transits to the next state $s_{t+1}$, which is sampled from the distribution $\mathcal{T}(\cdot|s_t, a_t)$. This process continues until a termination condition is met, which will be triggered within $H$ steps.

**Deterministic MDP for LLMs.** In text generation scenarios, the transition kernel $\mathcal{T}$ is deterministic, as each new state is formed by concatenating the previous tokens with the current output. The length limit for LLM outputs is characterized by $H$. Initially, an instruction $x$ is sampled from an initial distribution $\rho$. Each subsequent state $s_t = (x, a_{1:t-1})$ comprises the instruction $x$ and text pieces previously generated (e.g. reasoning steps in reasoning tasks). Each action $a_t$ is a text piece generated based on the current state $s_t$. The LLM policy $\pi(a_t|x, a_{1:t-1})$ maps the observed state to a distribution over the action space. The process reward model, $r : \mathcal{S} \times \mathcal{A} \rightarrow \mathbb{R}$, intuitively scores each action $a_t$ based on the instruction $x$ and previous generations $a_{1:t-1}$. For simplicity, the instruction $x$ is omitted in the state notation $(x, a_{1:t})$ thereafter when no ambiguity arises.

### 3.2 Defining $Q$-function Implicitly Defines a Reward Function

Recall that the state-action value $Q(s, a)$ (Mnih et al., 2013; Fan et al., 2020; Setlur et al., 2024) typically represents the expected benefit of taking a specific action $a$ to achieve a correct answer. In the context of reasoning tasks, we define the $Q$-value function as the success probability of achieving the correct final answer. Specifically, the $Q$-value function is defined as

$$\mathcal{Q}^\pi(a_{1:t-1}, a_t) := \sigma^{-1}\Big(\mathbb{E}_{a_{t+1:H} \sim \pi(\cdot|a_{1:t})}\mathcal{I}(x, a_{1:H})\Big), \tag{2}$$

where $\pi$ is a policy, $H$ is the maximum step number, $\sigma$ is the sigmoid function and $\sigma^{-1}$ is its inverse function to ensure $\mathcal{Q} \in \mathbb{R}$. $\mathcal{I}$ is an indicator function, which equals 1 if the trajectory reaches the correct answer of $x$, and 0 otherwise. For simplicity, we also denote $\mathcal{Q}(a_{1:t-1}, a_t)$ as $\mathcal{Q}_t$ when there is no ambiguity.

**Lemma 3.1.** *(Ng et al., 1999) For two reward functions $r(s_t, a_t)$ and $r'(s_t, a_t)$, if there exists a potential function $\Phi(s)$ satisfying $r'(s_t, a_t) = r(s_t, a_t) + \Phi(s_{t+1}) - \Phi(s_t)$, these two reward functions results in the same optimal policy.*

Given this lemma, defining the $Q$-value function implicitly defines a corresponding reward function.

**Lemma 3.2.** *Under deterministic MDP, the advantage function of the optimal policy $\pi^*$ can function the same as the reward function leading to $\pi^*$.*

*Proof.* Due to the deterministic MDP setting, we have $\mathcal{A}^*(s_t, a_t) = r(s_t, a_t) + V^*(s_{t+1}) - \mathcal{V}^*(s_t)$ where we denote the $Q, V$-value under the optimal policy $\pi^*$ as $Q^*, V^*$. Hence, with Lemma 3.1, we have the advantage function of the optimal policy functions the same as the reward function. $\square$

With the definition in Eq. 2, the advantage function of the optimal policy can be formulated as

$$\mathcal{A}^*(s_t, a_t) = \mathcal{Q}^*(s_t, a_t) - \mathbb{E}_{a_t \sim \pi^*(\cdot|s_t)}\mathcal{Q}^*(s_t, a_t) = \mathcal{Q}^*(s_t, a_t) - \mathcal{Q}^*(s_{t-1}, a_{t-1})$$

Thus, our objective is to approximate the $Q$-function of the optimal policy. However, the optimal policy is not known in advance and varies across different algorithms. To establish the relationships between $Q$-values at intermediate steps, we introduce the following mild assumption regarding ideal optimal policies.

**Assumption 3.1.** *For an ideal optimal policy $\pi^*$, the next step based on a correct state is more likely to be correct than be wrong, i.e. $\mathcal{P}^*(a_{t+1}|a_{1:t}) \gg \mathcal{P}^*(\overline{a_{t+1}}|a_{1:t})$, which follows achieving correct answer from a correct state is much easier than from a wrong state, i.e. $\mathcal{P}^*(\tau|s) > \mathcal{P}^*(\tau|\bar{s})$.*

In Section 4.3, we will further empirically validate this assumption. For the parameter notations of $\mathcal{P}^\pi(\cdot)$, we use the original notations to represent the correctness of states and an overline to indicate incorrectness. For example, $\mathcal{P}^\pi(\overline{a_{t+1}}|a_{1:t})$ denotes the probability that policy $\pi$ will produce an incorrect next step given the correct state sequence $a_{1:t}$, and $\mathcal{P}^\pi(\tau|\bar{s})$ represents the probability that the policy $\pi$ generates a correct trajectory from an incorrect state $s$. $\mathcal{P}^*$ is shorthand for $\mathcal{P}^{\pi^*}$, where $\pi^*$ is the optimal policy. Using the above definitions and assumptions, we can collect comparative labels to approximate $Q$-values, which will be introduced next.

### 3.3 Optimal $Q$-Value Ranking

In this subsection, we derive the $Q$-value rankings among intermediate reasoning steps. In our main Theorem 3.5, we establish that $Q$-values ascend with the continuation of correct steps and descend

as wrong steps proceed, while maintaining a significant gap between correct and wrong steps. To arrive at this result, we first derive the pairwise relationship between $Q$-values of an earlier step and a later step in Lemma 3.3. Next we show the relationship between the first correct step and the first incorrect step in Lemma 3.4. Finally, we combine these intermediate relationships to derive an integrated ranking across the entire trajectory.

We start by introducing a few lemmas that are useful for deriving our main Theorem 3.5. For any two actions $a_n, a_m$ s.t. $n < m$ in a solution trajectory $\tau = (x, a_1, a_2, ... a_H)$, we have

$$\mathcal{P}^*(\tau|a_{1:n}) = \mathcal{P}^*(a_{1:m}|a_{1:n})\mathcal{P}^*(\tau|a_{1:m}) + \mathcal{P}^*(\overline{a_{1:m}}|a_{1:n})\mathcal{P}^*(\tau|\overline{a_{1:m}}), \quad (3)$$

$$\mathcal{P}^*(\tau|\overline{a_{1:n}}) = \mathcal{P}^*(a_{1:m}|\overline{a_{1:n}})\mathcal{P}^*(\tau|a_{1:m}) + \mathcal{P}^*(\overline{a_{1:m}}|\overline{a_{1:n}})\mathcal{P}^*(\tau|\overline{a_{1:m}}), \quad (4)$$

which directly follows the Bayesian factorization. $\mathcal{P}^*(a_{1:m}|\overline{a_{1:n}})$ denotes the possibility that policy generate correct state $a_{1:m}$ conditioned on a wrong state $\overline{a_{1:n}}$. For a solution $\tau = (x, a_1, a_2, ... a_H)$, recall the $Q$ function in Eq.2, we define $\mathcal{Q}_\sigma^*(a_{1:t-1}, a_t) = \sigma(\mathcal{Q}^*(a_{1:t-1}, a_t)) = \mathcal{P}^*(\tau|a_{1:t})$ where $\sigma$ is the sigmoid function. Since $\sigma$ is a monotonically increasing function, hence when $\mathcal{Q}_\sigma^*(a_{1:m-1}, a_m) > \mathcal{Q}_\sigma^*(a_{1:n-1}, a_n)$ for any two steps $a_m, a_n$, we have $\mathcal{Q}^*(a_{1:m-1}, a_m) > \mathcal{Q}^*(a_{1:n-1}, a_n)$. Then we can obtain the following lemma.

**Lemma 3.3.** *For two steps $a_n, a_m$ in a solution $\tau$ where $n < m$, if they are both correct, we have $\mathcal{Q}^*(a_{1:n-1}, a_n) < \mathcal{Q}^*(a_{1:m-1}, a_m)$. If $a_n, a_m$ are both wrong, we have $\mathcal{Q}^*(a_{1:n-1}, a_n) > \mathcal{Q}^*(a_{1:m-1}, a_m)$.*

*Proof.* We first analyze the difference between the two correct steps as follows,

$$\mathcal{Q}_\sigma^*(a_{1:n-1}, a_n) - \mathcal{Q}_\sigma^*(a_{1:m-1}, a_m)$$
$$= \mathcal{P}^*(a_m|a_{1:n})\mathcal{P}^*(\tau|a_{1:m}) + \mathcal{P}^*(\overline{a_m}|a_{1:n})\mathcal{P}^*(\tau|\overline{a_{1:m}}) - \mathcal{P}^*(\tau|a_{1:m})$$
$$= \mathcal{P}^*(\overline{a_m}|a_{1:n})[\mathcal{P}^*(\tau|\overline{a_{1:m}}) - \mathcal{P}^*(\tau|a_{1:m})], \quad (5)$$

where the first equation uses the $Q$-function definition and Eq. 4, the second equation uses $\mathcal{P}^*(a_m|a_{1:n}) + \mathcal{P}^*(\overline{a_m}|a_{1:n}) = 1$. With the Assumption 3.1, we have $\mathcal{P}^*(\tau|\overline{a_{1:m}}) - \mathcal{P}^*(\tau|a_{1:m}) < 0$. Hence, when $a_n$ and $a_m$ are both correct, we have $\mathcal{Q}^*(a_{1:n-1}, a_n) < \mathcal{Q}^*(a_{1:m-1}, a_m)$. Similar to the above proof, we can factorize the $Q$-value difference between two incorrect steps as follows,

$$\mathcal{Q}_\sigma^*(a_{1:n-1}, a_n) - \mathcal{Q}_\sigma^*(a_{1:m-1}, a_m) = \mathcal{P}^*(a_m|\overline{a_{1:n}})[\mathcal{P}^*(\tau|a_{1:m}) - \mathcal{P}^*(\tau|\overline{a_{1:m}})]. \quad (6)$$

With the Assumption 3.1 where $\mathcal{P}^*(\tau|a_{1:m}) > \mathcal{P}^*(\tau|\overline{a_{1:m}})$, if $a_n, a_m$ are both incorrect, we have $\mathcal{Q}^*(a_{1:n-1}, a_n) > \mathcal{Q}^*(a_{1:m-1}, a_m)$. $\square$

Additionally, considering the initial situation intermediate steps and $\mathcal{V}^*(x)$, we have the following lemma.

**Lemma 3.4.** *For the first correct step $a_n$ and the first incorrect step $a_m$, we have $\mathcal{Q}^*(a_{1:n-1}, a_n) > \mathcal{V}^*(x) \gg \mathcal{Q}^*(a_{1:m-1}, a_m)$.*

*Proofs.* Considering the first correct step $a_n$, similar to the proof in Lemma 3.3, we have

$$\mathcal{Q}_\sigma^*(a_{1:n-1}, a_n) - \mathcal{V}_\sigma^*(x) = \mathcal{P}^*(\tau|a_{1:n}) - \mathcal{P}^*(\tau|x) = \mathcal{P}^*(\overline{a_n}|x)(\mathcal{P}^*(\tau|a_{1:n}) - \mathcal{P}^*(\tau|\overline{a_{1:n}})) \quad (7)$$

$$\mathcal{Q}_\sigma^*(a_{1:m-1}, a_m) - \mathcal{V}_\sigma^*(x) = \mathcal{P}^*(\tau|\overline{a_{1:m}}) - \mathcal{P}^*(\tau|x) = \mathcal{P}^*(a_m|x)(\mathcal{P}^*(\tau|\overline{a_{1:m}}) - \mathcal{P}^*(\tau|a_{1:m})) \quad (8)$$

Hence, we have $\mathcal{Q}^*(a_{1:m-1}, a_m) < \mathcal{V}^*(x) < \mathcal{Q}^*(a_{1:n-1}, a_n)$. Now, we obtain the ordering of the $Q$-value difference, but the specific discrepancy between intermediate steps has not been discussed yet. With Assumption 3.1, for an ideal $\pi^*$, we have $\mathcal{P}^*(\overline{a_n}|x) \ll \mathcal{P}^*(a_m|x)$. Hence, the difference between $\mathcal{V}^*(x)$ and the $Q$-value of the first correct step is much smaller than the difference between $\mathcal{V}^*(x)$ and the $Q$-value of the first incorrect step. $\square$

Based on the above derivations, we can rank the state-action $Q$-values for the whole trajectory. We formalize the ranking in the following theorem.

**Theorem 3.5** ($Q$-value ranking among reasoning steps)**.** *Formally, for a trajectory $\tau$ with $H$ steps, $C = [c_1, c_2, \ldots, c_{|C|}]$ denotes the index list of the **c**orrect intermediate steps, where $c_1 < c_2 < \cdots < c_{|C|}$, $W = [w_1, w_2, \ldots, w_{|W|}]$ denotes the index list of the **w**rong intermediate steps, where $w_1 < w_2 < \cdots < w_{|W|}$, we have*

$$\mathcal{Q}_{w_{|W|}}^* < \cdots < \mathcal{Q}_{w_2}^* < \mathcal{Q}_{w_1}^* \ll \mathcal{Q}_0^* < \mathcal{Q}_{c_1}^* < \mathcal{Q}_{c_2}^* < \cdots < \mathcal{Q}_{c_{|C|}}^*,$$

*where $Q_0^* = V^*(x)$, $|\cdot|$ denotes the length of the list, and $|C| + |W| = H$.*

### 3.4 COMPARATIVE LOSS FUNCTION FOR OPTIMIZING $Q$-VALUE RANKINGS

Given the optimal $Q$-value ranking derived in Theorem 3.5, we now propose a new comparative loss that trains RPM to approximate the intermediate $Q$-values. While the ranking relationship can be captured by the classical Plackett-Luce (PL) ranking model (Plackett, 1975; Luce, 1959), there are significant limitations when using the canonical PL loss directly in this context. The standard PL loss is designed to handle general ranking scenarios without accounting for the varying degrees of discrepancy within the ranking. However, in our case, the $Q$-value gaps between correct and incorrect steps are often highly pronounced (*cf.* Lemma 3.4), leading to a situation where the standard PL model may not adequately capture the importance of these differences. As discussed in Section 4.3, this results in suboptimal performance, since the PL loss does not differentiate sufficiently between steps that are only marginally different in rank versus those with substantial $Q$-value gaps.

**Comparative loss with $Q$-value margin.** To address the limitation, we adapt the vanilla PL loss to better reflect these discrepancies. Our proposed loss function is designed to emphasize the significant gaps in $Q$-values, ensuring that the model learns to prioritize these differences in a theoretically justified manner. The loss is defined as:

$$\mathcal{L}_{\text{theorem}} = -\frac{1}{H}\Big[\sum_{t=2}^{|W|} \log \frac{\exp(\mathcal{Q}_{w_t})}{\sum_{q=1}^{t} \exp \mathcal{Q}_{w_q}} + \sum_{t=0}^{|C|} \log \frac{\exp(\mathcal{Q}_{c_t})}{\sum_{q=0}^{t} \exp \mathcal{Q}_{c_q} + \sum_{w\in W} \exp(\mathcal{Q}_w + \zeta)}\Big], \quad (9)$$

where $\zeta$ is a margin hyperparameter introduced to emphasize the gap between correct and incorrect steps, and 0 is inserted at the beginning of $C$ for clarity.

Practically, prior research (Wang et al., 2023a; Shao et al., 2024) often treats all steps following the first incorrect step as wrong. Specifically, for a given trajectory $\tau = \{a_1, \ldots, a_{l-1}, a_l, \ldots, a_H\}$ where $a_{1:l-1}$ are correct steps and $a_l$ is the first incorrect step, existing data corpora typically categorize all subsequent steps $a_{l:H}$ as incorrect. This approach leads to a situation where the wrong steps are not necessarily accurately annotated, as they are all uniformly marked as incorrect. To address this issue and explore a practically effective loss function, we investigate several variations of the comparative loss function. Our practical implementation, which will be discussed in Section 4.3, is designed to better handle this scenario. The proposed loss function is:

$$\mathcal{L} = -\frac{1}{|C|}\sum_{t=0}^{|C|} \log \frac{\exp(\mathcal{Q}_{c_t})}{\sum_{q=0}^{t} \exp \mathcal{Q}_{c_q} + \sum_{w\in W} \exp(\mathcal{Q}_w + \zeta)}. \quad (10)$$

In this formulation, $\zeta$ is a positive scalar that adjusts the relative importance of incorrect steps, and $Q_0$ is set to 0 to simplify the computation. Comparing to $\mathcal{L}_{\text{theorem}}$, this objective disregards the internal rankings among incorrect steps, focusing solely on the relative rankings among correct steps and the substantial discrepancy between the $Q$-values of correct and incorrect steps, i.e. $\{\mathcal{Q}^*_{w_{|W|}}, \ldots, \mathcal{Q}^*_{w_2}, \mathcal{Q}^*_{w_1}\} \ll \mathcal{Q}^*_0 < \mathcal{Q}^*_{c_1} < \mathcal{Q}^*_{c_2} < \cdots < \mathcal{Q}^*_{c_{|C|}}$. We will perform extensive ablation comparing $\mathcal{L}$ and $\mathcal{L}_{\text{theorem}}$ in Section 4.3.

### 3.5 CLASSIFICATION-BASED PRM IS A SPECIAL CASE OF $Q$-VALUE APPROXIMATORS

We show that the previous classification-based PRM can be cast as a special case of our framework under certain conditions. To illustrate this, consider an extreme scenario where the assumptions outlined in Assumption 3.1 are satisfied, namely, when $\mathcal{P}^*(a_{t+1}|a_{1:t}) \to 1$ and $\mathcal{P}^*(\overline{a_{t+1}}|\overline{a_{1:t}}) \to 1$. According to the $Q$-function definition provided in Eq. 2 and leveraging Bayesian Factorization, it follows that classification-based PRMs approximate $Q$-value rankings under these conditions.

**Lemma 3.6.** *Formally, when $\mathcal{P}^*(a_{t+1}|a_{1:t}) \to 1$ and $\mathcal{P}^*(\overline{a_{t+1}}|\overline{a_{1:t}}) \to 1$ for any $t$, we have $\mathcal{Q}^*_\sigma(a_{1:m-1}, a_m) = 1$ for any correct step $a_m$ and $\mathcal{Q}^*_\sigma(a_{1:n-1}, a_n) = 0$ for any wrong step $a_n$.*

*Proof.* This result can be derived directly from Bayesian Factorization, which states:

$$\mathcal{P}^*(\tau|a_{1:m}) = \prod_{t=m+1}^{H} \mathcal{P}^*(a_t|a_{1:t-1}), \quad \mathcal{P}^*(\overline{\tau}|\overline{a_{1:n}}) = \prod_{t=n+1}^{H} \mathcal{P}^*(\overline{a_t}|\overline{a_{1:t-1}}). \quad (11)$$

Therefore, for a correct step, we have $\mathcal{Q}^*_\sigma(a_{1:m-1}, a_m) = \mathcal{P}^*(\tau|a_{1:m}) = 1$ and for a wrong step, we have $\mathcal{Q}^*_\sigma(a_{1:n-1}, a_n) = 1 - \mathcal{P}^*(\overline{\tau}|\overline{a_{1:n}}) = 0$. Thus, the cross-entropy loss used in classification-based PRMs can be interpreted as estimating the $Q$-value without bias. $\square$

| Sampling Policy | Methods | Dataset: MATH500 | | | | | Dataset: GSM-Plus | | | | |
|---|---|---|---|---|---|---|---|---|---|---|---|
| | | @8 | @16 | @32 | @64 | @128 | @8 | @16 | @32 | @64 | @128 |
| MetaMath-Mistral-7B | ORM | 32.8 | 34.8 | 36.2 | 39.0 | 38.2 | 56.58 | 57.63 | 57.17 | 57.63 | 58.33 |
| | $MSE_{1\text{-}0}$ | 33.2 | 36.2 | 37.6 | 38.8 | 38.4 | 58.21 | 58.75 | 58.71 | 58.50 | 58.17 |
| | $MSE_{MCTS}$ | 24.2 | 25.2 | 26.4 | 25.0 | 27.0 | 50.91 | 51.67 | 50.08 | 49.58 | 49.79 |
| | BCE | 33.6 | _37.0_ | 39.2 | 40.8 | _42.0_ | 59.25 | 60.29 | 61.16 | 61.88 | 61.72 |
| | PQM $\zeta = 2$ | _34.8_ | _37.0_ | _39.6_ | _41.8_ | 41.2 | **62.42** | **64.04** | **64.92** | **65.25** | **66.00** |
| | PQM $\zeta = 4$ | **36.2** | **38.2** | **41.0** | **44.2** | **44.6** | _62.04_ | _63.58_ | _64.50_ | _64.96_ | _65.20_ |
| Muggle-Math-13B | ORM | 24.0 | 28.0 | 27.0 | 28.8 | 28.2 | 55.41 | 55.83 | 56.83 | 54.83 | 54.45 |
| | $MSE_{1\text{-}0}$ | 28.2 | 30.2 | 33.0 | 33.6 | 34.0 | 56.42 | 58.42 | 58.38 | 58.67 | 59.08 |
| | $MSE_{MCTS}$ | 21.2 | 24.2 | 22.0 | 23.8 | 26.8 | 42.75 | 45.83 | 46.95 | 45.67 | 46.33 |
| | BCE | **30.4** | 31.4 | 33.4 | 36.4 | _37.0_ | 57.50 | 59.79 | 61.16 | 62.00 | 62.17 |
| | PQM $\zeta = 2$ | _30.0_ | _33.4_ | _34.4_ | _36.8_ | 35.0 | _60.58_ | _62.54_ | **64.25** | **64.79** | **65.62** |
| | PQM $\zeta = 4$ | _30.0_ | **34.8** | **36.2** | **39.2** | **39.0** | **61.00** | **62.66** | _64.08_ | **64.79** | _65.54_ |
| Llama-3-70B-Instruct | ORM | 45.0 | 46.0 | 43.4 | 42.4 | 43.2 | 71.66 | 71.50 | 72.00 | 71.66 | 71.13 |
| | $MSE_{1\text{-}0}$ | 41.6 | 42.2 | 40.0 | 36.8 | 38.0 | 71.79 | 71.67 | 71.96 | 71.25 | 71.04 |
| | $MSE_{MCTS}$ | 39.6 | 40.4 | 40.0 | 41.2 | 41.4 | 68.46 | 69.70 | 67.79 | 71.13 | 70.66 |
| | BCE | 43.6 | 41.4 | 41.6 | 42.4 | 39.8 | _72.16_ | 71.83 | 72.04 | 71.38 | 70.75 |
| | PQM $\zeta = 2$ | **47.6** | **49.0** | **50.4** | **48.4** | **51.4** | 72.04 | _71.95_ | _72.70_ | _72.33_ | _72.33_ |
| | PQM $\zeta = 4$ | _47.2_ | _48.2_ | _50.0_ | 46.0 | _47.8_ | **72.54** | **73.25** | **73.38** | **72.79** | _71.96_ |

Table 1: **Main results** measured by best-of-$n$ (BON@$n$) accuracy. The BON@1 of MATH500 for MetaMath-Mistral-7B is 24.4, for MuggleMath-13B is 18.4, for Llama-3-70B-Instruct is 37.4. The BON@1 of GSM-Plus for MetaMath-Mistral-7B is 48.0, for MuggleMath-13B is 43.16, for Llama-3-70B-Instruct is 67.875. **Boldface** and underline indicate the best two results.

# 4 EXPERIMENTS

## 4.1 EXPERIMENTAL SETTINGS

**Datasets and metrics.** Following previous research (Wang et al., 2023a; Lightman et al., 2024; Luo et al., 2024), we evaluate PRMs based on their verification ability through best-of-$n$ sampling. The metric, BON@$n$, assesses the correctness of the most preferred trajectory selected by the PRM from $n$ candidates for each question. During the evaluation, the PRM first scores every step within each trajectory. Consistent with prior studies (Wang et al., 2023a), the final score of a trajectory is determined by the minimum score of its individual steps. The test corpus includes 128 solutions for each question from GSM-Plus (Li et al., 2024) and MATH500 (Hendrycks et al., 2021) datasets. These solutions are sampled from three policy models with strong performance in math tasks with different scales: MetaMath-Mistral-7B (Yu et al., 2024), MuggleMath-13B (Li et al., 2023a), Llama-3-70B-Instruct (AI@Meta, 2024). We utilize the existing off-shelf corpus, Math-Shepherd (Wang et al., 2023a), as our training corpus.

**Baselines and implementation details.** Consistent with prior works (Wang et al., 2023a; Lightman et al., 2024), we evaluate the performance of PRM by comparing it against the outcome reward model (ORM). We also compare our comparative loss with the BCE loss, which is employed in Math-Shepherd. Additionally, some research (Zhang et al., 2024a; Wang et al., 2024) adopt more strict MSE loss to minimize the distance between the predicted value and the label. We implement MSE loss with two versions: 0-1 label and iterative Monte Carlo Tree Search (MCTS) to estimate the continuous label for MSE loss as in Zhang et al. (2024a). For the model architecture, we adopt general reward model frameworks, incorporating a value head on top of the Deepseek-7B-base LLM (Shao et al., 2024). This value head projects the latent representation of the model into a scalar value, facilitating the evaluation of intermediate steps and trajectories. More detailed implementation information, including specific configurations and experimental setups, can be found in Appendix B.

## 4.2 MAIN RESULTS

**Verification performance across different policy models.** Experimental results are shown in Table 1. Our proposed PQM demonstrates significant performance improvements over all baselines. Firstly, PQM outperforms the outcome reward model, which is consistent with prior findings that process-based methods provide a more nuanced evaluation of intermediate steps. Moreover, when compared to classification-based PRM models using BCE or MSE loss, PQM shows a notable advantage. For example, when verifying solutions sampled from the Llama-3-70B-Instruct model,

| Backbone for PQM | MetaMath-Mistral-7B | | | | | MuggleMath-13B | | | | | Llama-3-70B-Instruct | | | | |
|---|---|---|---|---|---|---|---|---|---|---|---|---|---|---|---|
| | @8 | @16 | @32 | @64 | @128 | @8 | @16 | @32 | @64 | @128 | @8 | @16 | @32 | @64 | @128 |
| Deepseek-math-7b-base | 36.2 | 38.2 | 41.0 | 44.2 | 44.6 | 30.0 | 34.8 | 36.2 | 39.2 | 39.0 | 47.2 | 48.2 | 50.0 | 46.0 | 47.8 |
| Deepseek-math-7b-rl | 38.0 | 40.8 | 42.8 | 45.4 | 44.2 | 31.8 | 34.6 | 38.6 | 37.2 | 37.4 | 49.8 | 50.8 | 53.2 | 53.8 | 55.0 |
| Qwen2-math-1.5b | 31.4 | 32.8 | 34.6 | 33.8 | 33.2 | 25.4 | 28.2 | 30.4 | 35.2 | 32.4 | 41.2 | 39.2 | 40.0 | 40.2 | 39.4 |
| Qwen2-math-1.5b-inst | 38.6 | 41.2 | 43.8 | 46.4 | 47.6 | 30.6 | 34.2 | 37.6 | 40.6 | 41.4 | 50.8 | 49.4 | 50.0 | 49.6 | 51.0 |
| Metamath-7b | 30.4 | 32.8 | 32.8 | 31.2 | 33.8 | 26.2 | 30.6 | 29.6 | 30.2 | 30.0 | 42.0 | 44.8 | 45.4 | 44.8 | 44.0 |
| Metamath-13b | 32.6 | 32.4 | 33.4 | 33.6 | 34.2 | 29.4 | 30.6 | 31.4 | 31.8 | 31.4 | 45.0 | 45.2 | 45.0 | 46.8 | 45.8 |

Table 2: Results of PQM across six **different LLM backbones** on MATH500. $\zeta$ is set to 4.

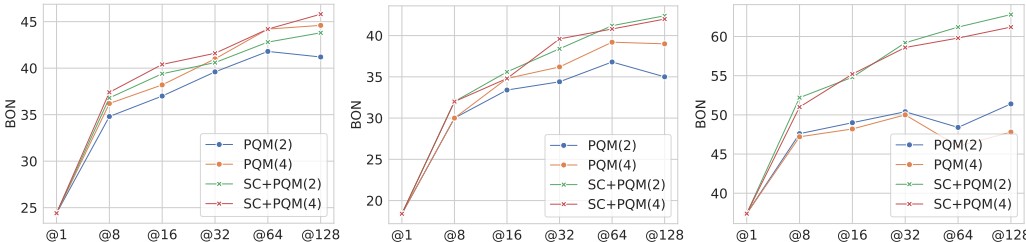

Figure 2: Integration of our approach PQM with self-consistency (SC) on three policy models, MetaMath-7B-Mistral (left), MuggleMath-13B (middle), Llama-3-70B-Instruct (right). The evaluation is conducted on MATH500. Numbers in brackets denote the value of $\zeta$.

PQM improves the accuracy from 39.8% (BCE) to 51.4%, a direct 11.6% improvement on the challenging MATH500 benchmark. This result underscores the effectiveness of PQM in capturing the relative quality of different steps within a trajectory, addressing the limitations of BCE loss which treats each step independently without considering their interdependencies. PQM outperforms MSE loss with either 0-1 label or MCTS search. Compared to 0-1 label, MCTS search requires more computational resources but only leads to marginal performance enhancement. This may stem from its $Q$-value definition with sophisticated heuristics, and theoretically biased estimation of $Q$-values in MCTS. Other results on both the MATH500 and GSM-Plus datasets across three policy models further confirm the efficacy of PQM. In these benchmarks, PQM consistently outperforms existing methods, demonstrating superior performance across different policy scales and test sets, validating the efficacy of ranking-based process reward modeling.

**PQM performance can be boosted by self-consistency (Wang et al., 2023b).** By sampling multiple trajectories and then selecting the final answer that appears most frequently, self-consistency can further enhance the reliability of LLMs. In Figure 2, we report performance when combining self-consistency with our method PQM under both $\zeta = 2$ and $\zeta = 4$. This integration capitalizes on the strengths of self-consistency to further enhance the verification. The performance gap between PQM and SC+PQM increases as we move to the right in Figure 2, since the large capacity model tends to reinforce the effectiveness of SC, leading to the increased performance gap observed in the figure. Our results reveal that this combination can boost performance, underscoring that blending self-consistency with process reward modeling provides a more effective verification strategy.

**PQM remains effective under different LLM backbones.** To explore the generalization of our approach, we train with PQM on additional LLM backbones, including Qwen2-Math-1.5B, Qwen2-Math-1.5B-Instruct (Yang et al., 2024), Deepseek-Math-7B-rl (Shao et al., 2024), Metamath-7B and Metamath-13B (Yu et al., 2024). As shown in Table 2, stronger backbones generally lead to better overall performance under the same sampling policy model. Moreover, Qwen2-Math-1.5B-Instruct achieves impressive results among six backbones, which indicates that a small-scale PQM can also provide effective verification if the backbone is specialized in mathematics.

### 4.3 FURTHER STUDIES

In ablation studies, we keep most of the experimental settings consistent with the main experiments, except that we use data with a length of less than 512 tokens, totaling 390k data out of 440k data, to save the training cost. The detailed hyperparameters are shown in Appendix B.

**Impact of margin $\zeta$.** In this ablation, we investigate how the margin $\zeta$ in our loss function influences the performance. We implement several variations with $\zeta = 0, 2, 4, 8, 16$. The experimental results are shown in Table 3, along with loss curves in Figure 5 (Appendix). Our experiments reveal that $\zeta$ has a minimal effect on the convergence of training, as the loss curves for all values flatten out

| Methods | MetaMath-Mistral-7B | | | | | MuggleMath-13B | | | | | Llama-3-70B-Instruct | | | | |
|---|---|---|---|---|---|---|---|---|---|---|---|---|---|---|---|
| | @8 | @16 | @32 | @64 | @128 | @8 | @16 | @32 | @64 | @128 | @8 | @16 | @32 | @64 | @128 |
| $\mathcal{L}, \zeta = 16$ | 34.6 | 36.4 | 38.2 | 40.2 | 39.2 | 29.6 | 32.4 | 34.6 | 35.4 | 35.0 | 42.4 | 43.6 | 40.2 | 40.2 | 39.0 |
| $\mathcal{L}, \zeta = 8$ | 36.4 | 40.2 | 41.2 | 43.8 | **44.6** | 30.8 | **33.8** | **37.2** | **38.8** | **38.8** | 47.0 | 47.0 | 47.8 | 46.2 | 46.0 |
| $\mathcal{L}, \zeta = 4$ | **36.8** | **40.6** | **41.8** | **44.4** | **44.6** | **32.0** | 33.6 | 36.8 | 38.4 | 37.4 | 47.4 | 47.0 | 45.6 | 47.8 | 48.2 |
| $\mathcal{L}, \zeta = 2$ | 35.8 | 39.0 | 40.8 | 43.4 | 43.8 | 30.2 | 32.8 | 34.2 | 36.8 | 37.4 | 47.4 | **49.0** | **50.6** | **51.2** | **50.4** |
| $\mathcal{L}, \zeta = 0$ | 32.8 | 37.0 | 36.2 | 35.8 | 36.4 | 26.2 | 27.4 | 29.2 | 29.2 | 28.0 | 44.6 | 44.4 | 45.4 | 44.2 | 46.6 |
| $\mathcal{L}_{\text{theorem}}, \zeta = 16$ | 33.2 | 34.6 | 35.0 | 37.2 | 38.0 | 28.8 | 30.6 | 32.4 | 32.6 | 32.6 | 46.2 | 45.4 | 44.8 | 44.8 | 44.2 |
| $\mathcal{L}_{\text{theorem}}, \zeta = 8$ | 33.6 | 34.4 | 35.0 | 35.4 | 35.6 | 29.0 | 29.4 | 30.0 | 31.4 | 32.6 | 43.8 | 42.6 | 41.0 | 38.2 | 37.4 |
| $\mathcal{L}_{\text{theorem}}, \zeta = 4$ | 35.4 | 38.2 | 39.0 | 40.0 | 40.2 | 31.6 | 33.2 | 34.8 | 36.4 | 34.8 | 44.8 | 45.2 | 46.4 | 47.8 | 46.0 |
| $\mathcal{L}_{\text{theorem}}, \zeta = 2$ | 33.8 | 35.8 | 37.6 | 37.6 | 38.0 | 28.4 | 29.4 | 31.0 | 31.4 | 32.0 | 43.0 | 44.8 | 46.0 | 47.8 | 48.6 |
| $\mathcal{L}_{\text{theorem}}, \zeta = 0$ | 30.4 | 29.8 | 30.6 | 31.8 | 33.0 | 24.0 | 26.8 | 29.0 | 28.8 | 26.2 | 41.6 | 40.4 | 40.6 | 40.4 | 37.4 |

Table 3: **Ablation results**. The BON@1 of MATH500 for MetaMath-Mistral-7B is 24.4, for MuggleMath-13B is 18.4, for Llama-3-70B-Instruct is 37.4. $\mathcal{L}, \mathcal{L}_{\text{theorem}}$ refers to Eq.10 and Eq.9 respectively. **Boldface** and underline indicate the best two results.

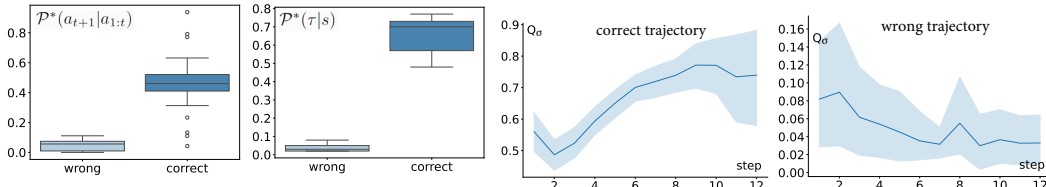

Figure 3: Empirical validation for Assumption 3.1.

Figure 4: Empirical evidence for Theorem 3.5.

after approximately 200 steps. However, the choice of $\zeta$ impacts the effectiveness of our method. As shown in Table 3, extreme values of $\zeta$—either too large or too small—lead to suboptimal performance. Specifically, $\zeta$ values of 2,4,8 yield the best results, whereas $\zeta$ values of 0 and 16 perform less effectively. When $\zeta$ is too large, the comparative loss overweighs the discrepancy between the correct steps and wrong steps while neglecting the ascending relationship among $Q$-values of correct steps. Conversely, when $\zeta$ is too small, the loss function fails to adequately capture $Q$-value discrepancies, leading to suboptimal performance. These findings align with our theoretical expectations and underscore the importance of choosing an appropriate $\zeta$ to balance the comparative loss and capture meaningful $Q$-value distinctions.

**Impact of loss design.** Since the empirical training dataset automatically marks all steps after the first incorrect one as negative steps, we ablate the impact of these pseudo-negative steps by comparing our loss function with the theoretical version as delineated in Eq. 9. The findings, presented in Table 3, reveal the existence of noise in negative annotations. Specifically, when applying the theoretical loss as in Eq. 9, there is a marked decline in performance. We also explored another variant that emphasize the first negative step since the first negative annotation is verified by the automatic annotation. The experimental results and analysis are supplemented in Appendix C.

**Empirical validation of Assumption 3.1 and Theorem 3.5.** To empirically validate the Assumption 3.1 and Theorem 3.5, we use Llama-3.1-70B-Instruct to substitute the optimal model $\pi^*$. We sample 256 trajectories from Math-Step-DPO-10K (Lai et al., 2024), each consisting of more than six steps. For each step $a_i$ in each trajectory, we sample 32 times by $\tau \sim \pi^*(\cdot|a_{1:i})$. In Fig. 3, the left panel's $y$-axis shows the proportion of correct next steps, while the right panel's $y$-axis displays the proportion of correct trajectories. The $x$-axis indicates whether the generation is conditioned on a correct state or an incorrect state. The plot demonstrates that when conditioned on a correct reasoning state, there is a higher probability of generating a correct subsequent step or completing a correct trajectory. This validates our Assumption 3.1. In Fig. 4, $x$-axis represents the $i$-th correct step (left) or wrong step (right), and $y$-axis represents the approximated $\mathcal{Q}_\sigma$. According to the graph, the approximated $Q$-values ascend with the continuation of the correct steps. Meanwhile, the latter wrong steps generally have smaller $Q$-values than the previous wrong steps. Moreover, there is a noticeable discrepancy between $Q$-value of correct steps (generally over 0.5) and incorrect steps (generally below 0.15). Implementation details and more discussions can be found in Appendix C.

**Qualitative example.** For each step in the solution, we display the predicted probability of achieving the correct final answer by ORM, classification-based PRM, and PQM in Table 4. We also show the original $\mathcal{Q}$ value predicted by PQM, along with $Q_\sigma = \sigma(Q)$. The $Q$-value predicted by PQM

| Q: Find all values of $x$ that satisfy the equation $x = \sqrt{11 - 2x} + 4$. | ORM | BCE | $\mathcal{Q}_\sigma$ | $\mathcal{Q}$ |
|---|---|---|---|---|
| Step 1: Subtract 4 from both sides of the equation. $x - 4 = \sqrt{11 - 2x}$ | - | 0.916 | 0.424 | -0.308 |
| Step 2: Square both sides of the equation. $(x - 4)^2 = (\sqrt{11 - 2x})^2$ | - | 0.882 | 0.487 | -0.053 |
| Step 3: Simplify. $x^2 - 8x + 16 = 11 - 2x$ | - | 0.848 | 0.482 | -0.070 |
| Step 4: Subtract 11 from both sides of the equation. $x^2 - 8x + 5 = 2x$ | - | 0.628 | 0.004 | -5.445 |
| Step 5: Subtract 2x from both sides of the equation. $x^2 - 10x + 5 = 0$ | - | 0.584 | 0.004 | -5.493 |
| Step 6: Factor the quadratic. $(x - 5)(x - 1) = 0$ | - | 0.489 | 0.002 | -6.164 |
| Step 7: The final answer is 5 and 1. I hope it is correct. | 0.475 | 0.399 | 0.001 | -6.811 |

Table 4: A case study on MATH500. The solution is sampled by Llama3-70B-Instruct. For each step, we display $Q$-value predicted by PQM($\mathcal{Q}$) and the estimated probability of achieving the correct answer by ORM, BCE, and our PQM($\mathcal{Q}_\sigma$). The steps after the first error (Step 4) are in gray.

has a sharp decrease at Step 4, which accurately locates the error. In contrast, the predicted probability of classification-based PRM only decreases smoothly and exhibits large values even for wrong steps. We show more qualitative examples in Appendix E.

## 5 RELATED WORKS

**Process Reward Models.** Process supervision (Uesato et al., 2022; Li et al., 2023b), represented by PRMs, can provide more precise feedback, which is easier for humans to interpret, and more directly rewards models in step-by-step reasoning tasks. Most existing research (Lightman et al., 2024; Wang et al., 2023a; Shao et al., 2024; Luo et al., 2024) formulates PRM as a classification problem, where the process reward is modeled as the probability of correctness of each step. We show that the prior approach can be cast as a special case under our theoretical framework. Due to the labor-intensive nature of dense annotations, several recent methods have introduced automatic annotation strategies (Wang et al., 2023a; Luo et al., 2024; Lu et al., 2024a). In these approaches, a step is deemed correct if a valid completion can be sampled from the LLM policy within $k$ trials, see details in Appendix A. Generally, the subsequent steps after the first error are all treated as wrong steps in this line of methods. Additionally, Zhang et al. (2024a); Wang et al. (2024) estimate the $Q$-value of intermediate steps by iterative Monte Carlo Tree Search (MCTS) and MSE loss. However, their $Q$-value designs are different from ours, which generally incorporate sophisticated heuristics, e.g., reasoning distance and quality value. Moreover, their works necessitate a dense online search over the large action space. Besides being costly, the distribution shift between the sampling policy and the optimal $\pi^*$ will result in biased estimation. In contrast, our comparative loss is easy to use, and can achieve unbiased estimation according to our theory. For completeness, we document the automatic annotation pipeline and more related research about PRM in Appendix A.

**MDP RL for LLMs.** Although the outcome reward model has advanced LLMs by applying reinforcement learning algorithms in bandit settings, it contradicts the auto-regressive nature of text generation and the step-by-step reasoning process. Recent studies (Rafailov et al., 2024a; Zhong et al., 2024; Xie et al., 2024; Zeng et al., 2024) introduced theoretically sound RL algorithms designed for LLMs in MDP settings. Although these efforts bridge the theoretical discrepancy in algorithms, they still rely, at least partially, on ORMs. Hence, the process reward model remains underexplored in MDP-based RL for LLMs. Orthogonal to our exploration, several works (Lu et al., 2024b; Lai et al., 2024; Chen et al.; Zhang et al., 2024b) adapt DPO (Rafailov et al., 2024b) to step-level preference optimization for reasoning tasks. We discuss the potential of integrating such methods into our framework in Appendix D.

## 6 CONCLUSION

In this paper, we introduce the Process Q-value Model (PQM), a new approach to model process rewards via optimization $Q$-value ranking. Unlike existing classification-based methods, which treat intermediate steps independently, PQM captures the interdependencies among steps. To effectively optimize the $Q$-value rankings, we propose a margin-based comparative training objective and validate its effectiveness through comprehensive experiments. Our results demonstrate that PQM significantly outperforms previous baselines, achieving an 11.6% accuracy improvement when verifying solutions generated by LLama-3-70B-Instruction on the MATH500 dataset, and consistently delivering robust results across various backbone scales, policy models, and datasets. We hope our work inspires more future investigation on process reward modeling that better captures the complexities of multi-step reasoning processes.

ACKNOWLEDGEMENT

We thank Leitian Tao and the ICLR reviewers for their valuable suggestions on the paper.

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

# A    RELATED WORKS

Several techniques have been developed to accelerate the data collection pipeline for training PRMs (Luo et al., 2024; Lu et al., 2024a). To simplify understanding, we first introduce the fundamental version proposed in Wang et al. (2023a). In this approach, the quality of an intermediate step is evaluated based on its potential to lead to the correct final answer. The pipeline can be summarized as follows:

- For a given question $x \sim \rho$, several trajectories are sampled by an LLM: $\tau_1, \cdots, \tau_N \sim \pi_1(\cdot|x)$. Each trajectory $\tau = \{a_1, a_2, \ldots, a_H\}$ consists of a sequence of steps, and the correctness of these steps is annotated through the following procedure.

- For a trajectory $\tau = \{a_1, a_2, \ldots, a_H\}$, we generate $n$ completions for each step from $a_1$ to $a_n$. Specifically, to annotate $a_i$, we sample $n$ completions by $\pi_2(\cdot|x, a_{1:i})$. The correctness of each completion is evaluated by final answer string matching.

- For each step $a_i$, if any completion of it achieves the correct final answer. We regard $a_i$ as correct, otherwise wrong. If $a_i$ is wrong, the subsequent steps $a_{i+1}, \cdots, a_n$ are all regarded as incorrect.

There have been several research trying to promote the pipeline efficiency. For example, Lu et al. (2024a) trains an additional confidence module to simplify the automatic annotations, Luo et al. (2024) performs a binary search to identify the first error location.

# B    IMPLEMENTATION DETAILS

All training is conducted on 8 NVIDIA A100-SXM4-80GB GPUs. We list the versions of the important external packages as follows: `torch==2.3.1`, `trl==0.8.0`, `flashattn==2.6.2`, `transformers==4.34.0`, `accelerate==0.33.0`, `deepspeed==0.13.1`, `nvidia-nccl-cu12==2.20.5`. We use the ZeRO-3 optimization stage of the deepspeed with bfloat16 precision. The hyperparameters for the ablation studies are provided in Table 5, and each training session for the ablation study took approximately 4.5 hours. For the main experiments, some training data has tokenized sequences longer than 2048 tokens, which limited the batch size and reduced training efficiency. To address this, we divide the training corpus into three groups based on tokenized length: sequences shorter than 512 tokens, between 512 and 1024 tokens, and greater than 1024 tokens. The batch sizes were set to 64, 24, and 8, respectively, for these groups. This strategy reduced the training time from about eleven hours to six hours. To generate the trajectories for Best-of-$n$ sampling, we use the VLLM pipeline with the temperature set to 1, top-p set to 1, and max length set to 2048. For the MCTS baseline, we fix the policy model as Qwen2-math-7B-Instruct, and utilize iterative MCTS search to train PRM. For a fair comparison, we use half of the Math-Shepherd corpus and its hard-estimated labels to construct $D_{V_0}$ (refer to the original paper (Zhang et al., 2024a)), and train an initial PRM. Then we conduct an MCTS search on questions of the remaining corpus. To keep the scale of the training set the same, we randomly sample trajectories with the quantity of 1/2 Math-Shepherd from the MCTS tree.

| hyper-parameter | value |
|---|---|
| scheduler | cosine |
| warm-up ratio | 0.1 |
| learning rate | 2e-6 |
| optimizer | AdamW |
| batch size per GPU | 64 |
| gradient accumulation steps | 4 |
| gradient checkpointing | True |

Table 5: Experimental settings for ablation studies.

| Methods | MetaMath-Mistral-7B | | | | | MuggleMath-13B | | | | | Llama-3-70B-Instruct | | | | |
|---|---|---|---|---|---|---|---|---|---|---|---|---|---|---|---|
| | @8 | @16 | @32 | @64 | @128 | @8 | @16 | @32 | @64 | @128 | @8 | @16 | @32 | @64 | @128 |
| $\mathcal{L}, \zeta = 8$ | 36.4 | 40.2 | 41.2 | 43.8 | 44.6 | 30.8 | 33.8 | 37.2 | 38.8 | 38.8 | 47.0 | 47.0 | 47.8 | 46.2 | 46.0 |
| $\mathcal{L}, \zeta = 4$ | 36.8 | 40.6 | 41.8 | 44.4 | 44.6 | 32.0 | 33.6 | 36.8 | 38.4 | 37.4 | 47.4 | 47.0 | 45.6 | 47.8 | 48.2 |
| $\mathcal{L}, \zeta = 2$ | 35.8 | 39.0 | 40.8 | 43.4 | 43.8 | 30.2 | 32.8 | 34.2 | 36.8 | 37.4 | 47.4 | 49.0 | 50.6 | 51.2 | 50.4 |
| $\mathcal{L}_{\text{ablate}}, \zeta = 8$ | 34.4 | 37.4 | 39.6 | 42.0 | 41.0 | 31.2 | 34.8 | 36.8 | 38.4 | 37.6 | 47.6 | 49.0 | 50.4 | 52.0 | 50.8 |
| $\mathcal{L}_{\text{ablate}}, \zeta = 4$ | 33.0 | 37.6 | 40.0 | 41.6 | 40.8 | 30.0 | 34.4 | 36.4 | 39.0 | 38.6 | 47.6 | 49.4 | 50.8 | 52.4 | 49.8 |
| $\mathcal{L}_{\text{ablate}}, \zeta = 2$ | 31.6 | 34.8 | 37.0 | 40.0 | 38.4 | 30.4 | 33.4 | 32.6 | 35.6 | 35.2 | 44.4 | 45.4 | 45.0 | 47.0 | 46.0 |
| $\mathcal{L}_{\text{ablate}}, \zeta = 0$ | 31.6 | 34.8 | 37.0 | 40.0 | 38.4 | 30.4 | 33.4 | 32.6 | 35.6 | 35.2 | 44.4 | 45.4 | 45.0 | 47.0 | 46.0 |

Table 6: **Ablation results**. The BON@1 of MATH500 for MetaMath-Mistral-7B is 24.4, for Llama-3-70B-Instruct is 37.4. $\mathcal{L}, \mathcal{L}_{\text{ablate}}$ refers to Eq.10, and Eq.12 respectively. The detailed hyperparameters for experiments of this table are shown in Appendix B.

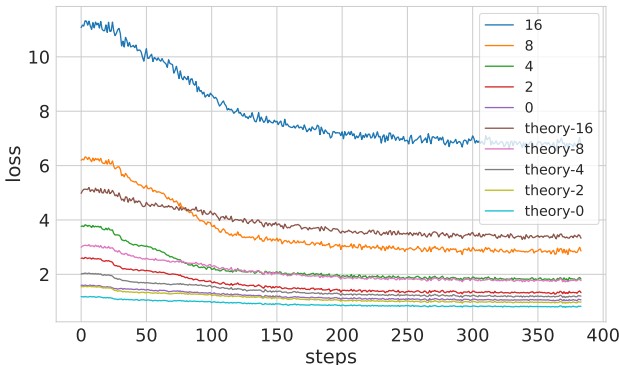

Figure 5: The loss curves for ablation studies in Table 3.

## C  ADDITIONAL EXPERIMENTS

**Loss variation.** Here, we explore what if we only emphasize the first incorrect step in the ranking. The loss variant is as follows,

$$\mathcal{L}_{\text{ablate}} = -\frac{1}{|C|} \sum_{t=0}^{|C|} \log \frac{\exp(\mathcal{Q}_{c_t})}{\sum_{q=0}^{t} \exp \mathcal{Q}_{c_q} + \exp(\mathcal{Q}_{w_1} + \zeta)}, \quad (12)$$

which promotes $\mathcal{Q}_{w_1}^* \ll \mathcal{Q}_0^* < \mathcal{Q}_{c_1}^* < \mathcal{Q}_{c_2}^* < \cdots < \mathcal{Q}_{c_{|C|}}^*$. As shown in Table 6, focusing only on the first negative step, which is verified by automatic annotation, the performance remains relatively stable, suggesting the limited utility of subsequent negative steps.

**Comparison with ceiling performance.** We evaluate the ceiling performance of various policy models and compare how PQM stands against this benchmark. Figure 6 presents the Pass@N metric alongside the best achievable verification performance for three distinct policy models. This comparison illustrates the upper limits of verification accuracy for each policy model and highlights the existing performance gaps. Specifically, the comparison suggests that current PRMs, including PQM, have not yet reached their full potential. These findings underscore the need for further advancements and refinements in PRM techniques to close the gap and approach the ceiling performance more closely.

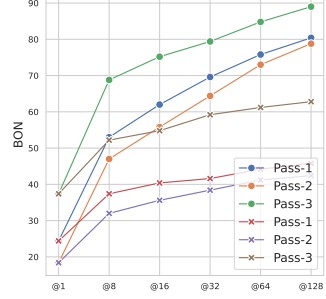

Figure 6: The ceiling performance and the best verification performance of three policy models on MATH500.

**Empirical validation for Assumption 3.1 and Theorem 3.5.** To empirically validate our Theorem 3.5, we use Llama-3.1-70B-Instruct to substitute the optimal model $\pi^*$. We sample 256 trajectories from Math-Step-DPO-10K (Lai et al., 2024), comprising 128 correct and 128 incorrect trajectories respectively. Each trajectory consists of more than six steps. If the reasoning state is included in a rejected answer, we

| Methods | Data Size | MetaMath-Mistral-7B | | | | | MuggleMath-13B | | | | | Llama-3-70B-Instruct | | | | |
|---------|-----------|------|------|------|------|------|------|------|------|------|------|------|------|------|------|------|
| | | @8 | @16 | @32 | @64 | @128 | @8 | @16 | @32 | @64 | @128 | @8 | @16 | @32 | @64 | @128 |
| BCE | 25% | 19.6 | 21.0 | 18.2 | 19.0 | 17.8 | 17.6 | 16.8 | 15.8 | 15.2 | 13.8 | 37.2 | 35.6 | 34.2 | 34.6 | 30.0 |
| | 50% | 23.6 | 24.2 | 22.8 | 22.4 | 19.8 | 17.2 | 17.8 | 17.0 | 14.2 | 13.0 | 37.6 | 35.4 | 32.6 | 31.8 | 29.0 |
| | 75% | 32.4 | 31.8 | 34.0 | 34.6 | 33.6 | 28.4 | 28.4 | 31.0 | 31.6 | 31.6 | 40.6 | 38.8 | 37.0 | 38.4 | 38.8 |
| | 100% | 33.6 | 37.0 | 39.2 | 40.8 | 42.0 | 30.4 | 31.4 | 33.4 | 36.4 | 37.0 | 43.6 | 41.4 | 41.6 | 42.4 | 39.8 |
| PQM | 25% | 21.4 | 21.6 | 19.8 | 19.8 | 19.2 | 18.0 | 15.4 | 17.0 | 14.8 | 14.0 | 37.4 | 36.6 | 37.2 | 38.4 | 35.6 |
| | 50% | 21.0 | 22.0 | 20.2 | 20.2 | 19.4 | 18.6 | 16.8 | 16.6 | 14.0 | 14.2 | 37.4 | 36.4 | 34.4 | 34.2 | 32.6 |
| | 75% | 33.4 | 36.4 | 37.0 | 39.6 | 38.0 | 29.2 | 32.4 | 35.0 | 37.2 | 37.4 | 46.8 | 47.8 | 47.0 | 47.2 | 46.0 |
| | 100% | 36.2 | 38.2 | 41.0 | 44.2 | 44.6 | 30.0 | 34.8 | 36.2 | 39.2 | 39.0 | 47.2 | 48.2 | 50.0 | 46.0 | 47.8 |

Table 9: The Best-of-$n$ performance of PRMs trained on different data size. The comparisons are conducted on classification-based PRM (BCE loss) and our PQM. The BON@1 of MATH500 for MetaMath-Mistral-7B is 24.4, for MuggleMath-13B is 18.4, for Llama-3-70B-Instruct is 37.4.

regard this reasoning state as incorrect. For each reasoning state $a_{1:i}$ in each trajectory, we sample 32 completions with $\tau \sim \pi^*(a_{1:i})$. The correctness of next-step $a_{i+1}$ is annotated automatically as in Wang et al. (2023a) with Qwen2-Math-Instruct-7B. We use statistics after the fifth step to avoid Qwen2-Math-Instruct-7B having a larger possibility to self-correct the step, hence misleading the label. We also count the correctness of each whole trajectory to approximate $\mathcal{Q}_\sigma$ for $a_{1:i}$ as defined in Eq. 2. In Fig. 4, we count the correctness proportionality of correct completions according to the position $i$ of the reasoning state $a_{1:i}$. According to the left subgraph of Fig. 4, the approximated $\mathcal{Q}_\sigma$ ascends with the continuation of the correct steps. The right subgraph illustrates that the latter wrong steps generally have smaller $Q$-values than the previous wrong steps. Moreover, there is a noticeable discrepancy between the $Q$-value of correct steps with $\mathcal{Q}_\sigma$ generally over 0.5 and incorrect steps with $\mathcal{Q}_\sigma$ generally below 0.15.

**PRM-guided beam search.** To further validate the effectiveness of our PQM, we have conducted additional experiments on PRM-guided beam search. The comparison is conducted between PQM and classification-based PRMs with BCE loss. We set the beam size as 8, and the generative temperature as 0.7. The evaluation is conducted on MATH500 across two policy models, Llama-3-8B-Instruct (AI@Meta, 2024) and Eurus-7b-sft (Yuan et al., 2024). The results are reported in Tabel 7, which demonstrate that PQM can more effectively guide the LLM to reason. For ablations, we compare the performance of PQMs trained with $\mathcal{L}_{\text{theorem}}$ and different $\zeta$ values as in §4.3. We use Eurus-7b-sft as the policy model. The results in Table 8 align with the findings from the Best-of-N experiments, showing that a sufficiently large range of $\zeta$ leads to strong performance in PRM-guided beam search, with optimal values typically falling in the middle of the range.

| Policy Models | Pass@1 | BCE | PQM |
|---------------|--------|------|------|
| Llama-3.1-8B-Instruct | 17.2 | 26.4 | 31.6 |
| Eurus-7b-sft | 19.4 | 24.2 | 29.2 |

Table 7: The performance of PRM-guided beam search on MATH500.

| Objective | $\zeta = 1$ | $\zeta = 2$ | $\zeta = 4$ | $\zeta = 8$ | $\zeta = 16$ |
|-----------|-------------|-------------|-------------|-------------|--------------|
| $\mathcal{L}$ | 26.4 | 27.8 | 28.8 | 28.4 | 25.6 |
| $\mathcal{L}_{\text{theorem}}$ | 24.8 | 26.0 | 28.0 | 28.2 | 26.6 |

Table 8: Ablation Studies of PQM-guided beam search on MATH500. The sample policy is Eurus-7b-sft.

**Sample-efficiency of PQM.** To examine whether PQM robustly outperforms classification-based PRM across different dataset sizes, we randomly sample $25\%, 50\%, 75\%$ of the original dataset to train PRMs with BCE loss and PQM loss. We keep most of the hyperparameters as in our main experiments, and set $\zeta$ as 4. As shown in Table 9, the results suggest that PQM generally outperforms BCE on all ranges of data sizes, and is more sample efficient.

**Comparison of ranking behaviors between PQM and BCE.** We first highlight behavioral differences based on the qualitative example in Table 4. **(1) BCE produces probabilities that are monotonically decreasing for correct steps** (step 1: $0.916 \rightarrow$ step 2: $0.882 \rightarrow$ step 3: $0.848$). This behavior contradicts the desired property established in Theorem 3.5, which proves that values should increase (rather than decrease) for correct reasoning steps. **(2) BCE does not produce a large transition in values between correct and incorrect steps.** For example, in Table 4, the prob-

ability only slightly decreases from 0.848 (step 3) to 0.628 (step 4), failing to sharply differentiate between correct and incorrect steps. In contrast, our PQM framework produces $Q$-values with a significant drop from correct to incorrect steps, better aligning with the desired behavior. For example, in Table 4, the $Q_\sigma$ value drops substantially from 0.482 to 0.004 between steps 3 and 4.

Statistically, we conduct an empirical study to confirm whether BCE and PQM result in different rankings on test steps. We calculate the proportion of solutions where classification-based PRM and PQM produce the same rankings across steps. In the test set, only 29.18% of solutions shared the same rankings. Furthermore, when comparing rankings across different solutions for the same question (Best-of-N results), we observed that 0% of test questions had identical rankings. We also randomly sample 2048 cases from the training set. Statistically, classification-based PRMs and PQM yield different ranking behaviors on 62.79% training cases. These statistics indicate a significant behavioral difference between BCE and PQM.

## D  INTER-SOLUTION COMPARISON

The comparison introduced in the main paper can be termed as intra-solution comparison, since two compared reasoning steps are within a single trajectory. This is partially because of the format of currently available corpora for PRM, which generally treats a single trajectory as a data point. Nevertheless, Theorem. 3.5 can seamlessly apply to comparison among different trajectories, i.e., inter-solution comparison. For instance, if two trajectories are diverged from $t$-th step with a common correct prior $a_{1:t-1}$, the comparison will proceed between two different $t$-th steps. Here, we denote $a_t^c$ is the correct one while $a_t^w$ is the wrong one. In this setting, we can derive the following corollary (note that $\mathcal{Q}$ represents the optimal Q-function $\mathcal{Q}^*$ if no ambiguity.

**Corollary D.1** ($Q$-value ranking for inter-solution comparison). *Formally, for two trajectories with the same correct prior $a_{1:t-1}$ and $a_t^c \succ a_t^w$, the Q-value rankings among these steps are as follows, $\mathcal{Q}_t^w \ll \mathcal{Q}_0 < \mathcal{Q}_1 < \cdots < \mathcal{Q}_{t-1} < \mathcal{Q}_t^c$, where $Q_0 = V(x)$.*

There have been several offline step-level DPO methods (Lu et al., 2024b; Lai et al., 2024; Chen et al.; Zhang et al., 2024b) concurrent to our research. Though not focused on PRM, their theoretical derivations can also be encompassed by the inter-solution comparison as in Corollary D.1. Moreover, they (Lai et al., 2024) generally only utilize $\mathcal{Q}_t^w \ll \mathcal{Q}_t^c$ and discard the ranking relationships among intermediate steps.

**Corollary D.2** ($Q$-value ranking for inter-solution comparison (General Version)). *Formally, for a trajectory $\tau$ with successive $H$ step pairs, $[(a_1^c, a_1^w), (a_2^c, a_2^w), \cdots, (a_H^c, a_H^w)]$, where the $t$-th step is conditioned on previous correct steps, i.e. $a_t \sim \pi(\cdot|a_{1:t-1}^c)$. We have $\mathcal{Q}_H^w < \cdots < \mathcal{Q}_2^w < \mathcal{Q}_1^w \ll \mathcal{Q}_0 < \mathcal{Q}_1^c < \cdots < \mathcal{Q}_{t-1}^c < \mathcal{Q}_H^c$ where $Q_0 = V(x)$.*

Some step-level optimization are based on tree-structured data as in Corollary D.2, however, existing research (Chen et al.; Zhang et al., 2024b) generally considers each step pair independently, i.e. $\mathcal{Q}_1^w < \mathcal{Q}_1^c, \cdots, \mathcal{Q}_H^w < \mathcal{Q}_H^c$, neglecting the integral $Q$-value rankings within the tree. We believe the above corollaries can shed some light on step-level offline optimization.

## E  CASE STUDIES

We show more qualatative examples in Table 10 and Table 11. We directly report $\mathcal{Q}^*$ to show the value transition. The predicted values generally have a slight increase among correct steps. When encountering the wrong step, the predicted value will have a sharp decrease.

## F  LIMITATIONS & FUTURE WORKS

Notably, as shown in Luo et al. (2024), the data quality of Math-shepherd is restricted due to the noise introduced by automatic annotation. We believe that more advanced datasets could significantly improve the verification capabilities of PRMs. As shown in Appendix C, there remains a considerable gap between the current performance of PRMs and the ceiling performance, underscoring the need for further advancements in PRM techniques. Additionally, while PRMs have garnered increasing research attention, there is a lack of systematic studies on corresponding online RL algorithms that leverage PRMs. The potential for fully utilizing PRMs in this context remains largely unexplored.

---

**Question:** What is the smallest positive perfect cube that can be written as the sum of three consecutive integers? Correct Answer: 27

---

**Step 1:** Do some research about cubes and about consecutive integers. The equation we are looking for is in the form of $x^3 = y + (y + 1) + (y + 2)$. (-1.703125)
 **Step 2:** Add $y + (y+1) + (y+2)$, the sum of the 3 consecutive integers.y+(y+1)+(y+2)=3y+3 (-0.97265625)
**Step 3:** Make the equations equal to each other.$x^3 = 3y + 3$ (-1.0625)
**Step 4:** Subtract 3 out of both sides.$x^3 - 3 = 3y$ (-0.796875)
**Step 5:** Divide both sides by 3.$\frac{x^3-3}{3} = y$ (-0.79296875)
**Step 6:** Since y has to be an integer, look for a perfect cube that, when subtracted by 3, it results in a multiple of 3.Our smallest option is to try 1, but it doesn't work.Our next option is to try 2, but it doesn't work.Our next option is to try 3, but it doesn't work.Our next option is to try 4, which results in 61.Our next option is to try 5, which results in 106. (-3.234375)
**Step 7:** Check which one works. 61 isn't a multiple of 3, but 106 is. Final Answer: The final answer is 125. I hope it is correct. (-5.03125)

---

**Step 1:** Write an expression for three consecutive integers. Call them $a - 1, a$, and $a + 1$. (-0.53515625)
**Step 2:** Write an expression for their sum. The sum of the three is $a - 1 + a + a + 1 = 3a$. (-0.302734375)
**Step 3:** We want this sum to be a perfect cube, so let $3a = n^3$. Divide both sides by 3 to get $a = \frac{n^3}{3}$. (-0.265625)
 **Step 4:** Since we want $a$ to be an integer, it must be that $n$ is a multiple of 3. Therefore, we have $n = 3k$, where $k$ is an integer. Substitute this back into $a = \frac{n^3}{3}$ to get $a = \frac{(3k)^3}{3} = 3^2 k^3$. (-0.33203125)
 **Step 5:** From here, we just guess and check for values of $k$ to find our answer. When $k = 1$, then $a = 3^2(1)^3 = 9$ and $a - 1 = 8$, $a = 9$, and $a + 1 = 10$. The sum of these three consecutive integers is $8 + 9 + 10 = 27 = 3^3$. Therefore, the smallest positive perfect cube that can be written as the sum of three consecutive integers is $\boxed{27}$. Final Answer: The final answer is 27. I hope it is correct. (-1.6015625)

---

**Step 1:** Let $n, n + 1, n + 2$ be three consecutive integers, such that their sum is equal to $k^3$, a perfect cube. In other words, $n + (n + 1) + (n + 2) = k^3$. (-0.734375)
 **Step 2:** Rearranging the terms in the equation, we obtain: $3n + 3 = k^3$. (-0.255859375)
**Step 3:** Substracting 3 from both sides, we get: $3n = k^3 - 3$. (-0.177734375)
**Step 4:** Dividing both sides by 3, we obtain: $n = \frac{k^3-3}{3}$. (-0.376953125)
**Step 5:** Since $n$ must be an integer, then $k^3 - 3$ must be divisible by 3. (-0.451171875)
**Step 6:** Adding 3 to both sides, we get: $k^3 = k^3 - 3 + 3, k^3 = k^3 - 3 + 3, k^3 = (k^3 - 3) + 3$. (-2.3125)
 **Step 7:** The smallest perfect cube that is 3 greater than a multiple of 3 is $k^3 = 64$. (-5.46875)
**Step 8:** Plugging in $k^3 = 64$ to our equation in Step 4, we get: $n = \frac{64-3}{3} = \frac{61}{3} = 20 + \frac{1}{3}$. (-5.71875)
**Step 9:** Since $n$ must be an integer, then we must increase $k^3$ to $k^3 = 125$ to get an integer value of $n$. (-5.0625)
 **Step 10:** Plugging in $k^3 = 125$ to our equation in Step 4, we get: $n = \frac{125-3}{3} = \frac{122}{3} = 40 + \frac{2}{3}$. (-5.34375)
 **Step 11:** Since $n$ must be an integer, then we must increase $k^3$ to $k^3 = 216$ to get an integer value of $n$. (-4.625)
**Step 12:** Plugging in $k^3 = 216$ to our equation in Step 4, we get: $n = \frac{216-3}{3} = \frac{213}{3} = 71$. (-4.0625)
**Step 13:** We found an integer value of $n$, which is our goal. We will not need to increase $k^3$ any further. (-1.8359375)
**Step 14:** Plugging in $n = 71$ to our initial equation, we get: $71+72+73 = k^3, 216 = k^3, k^3 = 216$. Final Answer: The smallest positive perfect cube that can be written as the sum of three consecutive integers is 216. I hope it is correct. (-4.84375)

---

Table 10: A case of MATH500 sampled by Llama-3-70B-Instruct. The red value represents the predicted $\mathcal{Q}^*$ provided by PQM.

**Question:** Find all values of $x$ that satisfy the equation $x = \sqrt{11 - 2x} + 4$. Correct Answer: 5

**Step 1:** Subtract 4 on both sides of the equation to isolate the radical.$x - 4 = \sqrt{11 - 2x}$ (-0.349609375)

**Step 2:** Square both sides to get rid of the radical.$(x - 4)^2 = 11 - 2x$ (-0.12353515625)

**Step 3:** Expand, combine like terms, and rewrite the equation in standard form (leading coefficient of 1) to have a quadratic in $x$. $x^2 - 10x + 21 = 0$,$x^2 - 7x - 3x + 21 = 0$, $x(x - 7) - 3(x - 7) = 0$,$(x - 7)(x - 3) = 0$ (-1.28125)

**Step 4:** Factor out the solutions to the equation.$x - 7 = 0$ or $x - 3 = 0$,$x = 7$ or $x = 3$ (-1.3984375)

**Step 5:** Check the solutions in the original equation to see if they satisfy the equation. When $x = 7$,$7 = \sqrt{11 - 14} + 4$,$7 = \sqrt{-3} + 4$ No real solution for $x = 7$. When $x = 3$,$3 = \sqrt{11 - 6} + 4$,$3 = \sqrt{5} + 4$,$3 - 4 = \sqrt{5}$,$-1 = \sqrt{5}$. No real solution for $x = 3$ either.Final Answer: The final answer is there is no solution. I hope it is correct. (-3.640625)

---

**Step 1:** Isolate the radical $x - 4 = \sqrt{11 - 2x}$. (-0.251953125)

**Step 2:** Square both sides $(x - 4)^2 = (\sqrt{11 - 2x})^2$. (-0.01080322265625)

**Step 3:** Simplify $x^2 - 8x + 16 = 11 - 2x$. (-0.022216796875)

**Step 4:** Write in standard form $x^2 - 6x + 5 = 0$. (0.0849609375)

**Step 5:** Factor $(x - 5)(x - 1) = 0$.. (0.1630859375)

**Step 6:** Solve for $x$, $\boxed{x = 1, 5}$. (-0.44140625)

**Step 7:** Check solutions: $x = 1$ is extraneous. Final Answer: The final answer is 5. I hope it is correct. (-1.40625)

---

**Step 1:** We want to first isolate the square root on the right hand side. Subtract 4 from both sides to get $x - 4 = \sqrt{11 - 2x}$. (0.0390625)

**Step 2:** Now we square both sides to get$(x - 4)^2 = 11 - 2x$. (0.09521484375)

**Step 3:** Multiply out to get$x^2 - 8x + 16 = 11 - 2x$.(0.2275390625)

**Step 4:** Move all terms to the left and combine like terms to get$x^2 - 10x + 5 = 0$.(-4.375)

**Step 5:** We now factor the equation to get $(x - 5)(x - 1) = 0$.(-4.5)

**Step 6:** And we solve for x by setting each factor equal to 0: $x = 1$ or $x = 5$. Final Answer: The final answer is $x = \boxed{1}$ or $x = \boxed{5}$. I hope it is correct. (-6.5)

---

**Step 1:** Subtract 4 from both sides of the equation.$x - 4 = \sqrt{11 - 2x}$ (-0.296875)

**Step 2:** Square both sides of the equation.$(x - 4)^2 = 11 - 2x$ (-0.0341796875)

**Step 3:** Expand and simplify both sides of the equation.$x^2 - 8x + 16 = 11 - 2x$,$x^2 - 8x + 2x - 16 + 11 = 0$,$x^2 - 6x - 5 = 0$,$(x - 5)(x + 1) = 0$ (-4.875)

**Step 4:** Solve for both values of $x$.$x - 5 = 0$,$x = \boxed{5}$,$x + 1 = 0$,$x = \boxed{-1}$ (-5.5)

**Step 5:** Check both answers in the original equation to ensure they are valid.Final Answer: The final answers are 5 and -1. I hope it is correct. (-6.375)

---

Table 11: Case studies on MATH500. Solutions are sampled from Llama-3-70B-Instruct. The red value represents the predicted $\mathcal{Q}^*$ provided by PQM.

