# OpenReview forum: "Process Reward Model with Q-value Rankings"
_ICLR.cc/2025/Conference — ICLR 2025 Poster_

### Official Review · Reviewer_1S6Q · 2024-10-16

**Soundness:** 3
**Presentation:** 2
**Contribution:** 2
**Rating:** 3
**Confidence:** 4

**Summary:**

This paper introduces PQM as an improvement over existing PRM methods, particularly in tasks requiring complex reasoning and decision-making. Traditional PRMs, typically framed as classification problems using cross-entropy loss, evaluate each step independently, which can result in suboptimal reward allocation and fails to account for interdependencies between steps. PQM redefines PRM using a MDP framework, optimizing Q-value rankings through a novel comparative loss function that better captures the dynamics of sequential decision-making. The authors claim that PQM offers a more detailed and theoretically robust method for distributing rewards across a process. Empirical evaluations demonstrate that PQM outperforms classification-based PRMs across different language model backbones, sampling policies, and multi-step reasoning tasks. Additionally, ablation studies confirm the effectiveness of the comparative loss function. The code is made available for reproducibility.

**Strengths:**

PQM presents clear and well-structured experimental results, providing strong empirical evidence of its effectiveness across various benchmarks. The thorough evaluations, including comparisons with classification-based PRMs and comprehensive ablation studies, effectively demonstrate the advantages of the proposed framework and its comparative loss function.

**Weaknesses:**

The paper lacks clarity regarding the necessity of the OSF framework. It does not sufficiently justify why this specific approach is needed to address the challenges in reinforcement learning, or how it fundamentally improves upon existing counterfactual explanation methods. The rationale behind why the OSF state is more effective than traditional approaches remains underexplained, making it difficult to assess its true impact.

**Questions:**

N/A

---

> ### Author Response · Authors · 2024-11-16
> **Responses to Reviewer 1S6Q**
>
> Dear Reviewer 1S6Q,
>
> Thank you for taking the time to review our submission. We noticed that your comments mention the "OSF framework," which does not appear in our paper, as well as points related to reinforcement learning algorithms and counterfactual explanation methods.
>
> It seems **there may have been a mix-up with feedback meant for another paper.** Could you please kindly review the comments and let us know if any adjustments might be needed to reflect our paper’s content more accurately?
>
> Thank you very much for your efforts in providing feedback on our work.
>
> Sincerely,
>
> Authors

---

> > ### Comment · Reviewer_1S6Q · 2024-11-26
> > **No change.**
> >
> > Same decision.

---

> ### Author Response · Authors · 2024-11-19
>
> Dear Reviewers 1S6Q:
>
> We sincerely appreciate your time and effort in reviewing our paper. We are following up as it appears that **part of your review may have been accidentally intended for a different submission OSF**.
>
> Before the author-reviewer discussion phase concludes, we would greatly value the opportunity to address any concerns or questions related to our work. We are eager to engage in a constructive discussion with you to ensure an accurate assessment.
>
> Thank you again for your understanding and feedback.
>
> Sincerely,
>
> Authors

---

### Official Review · Reviewer_qfhZ · 2024-11-01

**Soundness:** 4
**Presentation:** 4
**Contribution:** 3
**Rating:** 8
**Confidence:** 3

**Summary:**

This paper proposes a new reward modeling framework for LLMs by using Q-value ranking to enhance the credit assignment of each state’s contribution to the outcome. Both theoretical and empirical results demonstrate the effectiveness of the proposed framework.

**Strengths:**

- In general, the paper is well-written and the proposed framework is easy to follow and understand.
- The theoretical results support the use of Q-value ranking, which is further validated by empirical results.
- The relationship between the proposed framework and prior work (classification-based PRM) is compelling.
- The experimental results are comprehensive, and the case study helps readers easily grasp the concepts.

**Weaknesses:**

It is hard to find any weaknesses.

**Questions:**

- Are there cases where Q-values are inaccurately trained?
- How does inaccuracy in Q-values impact performance?
- It would be beneficial if the authors provided a discussion on inaccuracies in Q-values, as this could offer an opportunity to improve the proposed method.

---

> ### Author Response · Authors · 2024-11-16
> **Response to Reviewer qfhZ**
>
> We thank the reviewer for the positive and kind comments! We are deeply encouraged the reviewer recognized the strengths of our work from various perspectives. Below we respond to your questions in detail.
>
>
> > Inaccuracies of Q-values. (Are there cases where Q-values are inaccurately trained? How does inaccuracy in Q-values impact performance?)
>
> Thank you for these insightful questions. In practice, $Q$-values may be inaccurately learned due to noise or annotation errors of training corpora. For instance, in the existing training corpus for PRM, e.g. Math-Shepherd, steps following an initial mistake are automatically labeled as incorrect, which introduces pseudo-negative labels that can affect ranking accuracy of $Q$-values. These inaccuracies can lead the model to misinterpret the relative importance of intermediate steps, impacting its ability to distinguish correct from incorrect reasoning paths.
>
> To mitigate this, we have adapted our theoretical loss to a practical version that accounts for this annotation noise, as discussed in Lines 294-303. We believe a more refined data collection process can further enhance the performance of PQM, but it falls outside the primary scope and focus of our paper, which centers on the learning mechanism itself. This is indeed a significant and valuable question that future research could follow.

---

### Official Review · Reviewer_QeSG · 2024-11-02

**Soundness:** 3
**Presentation:** 3
**Contribution:** 4
**Rating:** 8
**Confidence:** 3

**Summary:**

In the presented work, the authors introduce a novel approach to process reward modeling that focuses on the contribution of intermediate reasoning steps and their respective contributions to the final methods. In contrast to existing PRM approaches, the authors introduce a Q-value base method (Process Q-value Model) that is better suited for the problem by using a ranking approach, thereby capturing inter-dependencies among intermediate reasoning steps. The problem addressed by the authors is relevant to the current field, particularly for tasks requiring multi-step reasoning, like MATH500, and is thereby timely and demonstrates significant performance improvements over prior work.

**Strengths:**

- Sections 2 and 3 are well written and provide sufficient context of the field as well as the proposed method
- The experiments are comprehensive and, without a doubt, the proposed method achieves high performance across multiple LLM backends and math benchmarks, making it a strong contribution.

**Weaknesses:**

- Section 3 could benefit from some intuition. While the content seems sound, some higher-level guidance could be beneficial (this is just a minor issue)
- Experiments on self-consistency could benefit from some additional explanations. In particular, given figure-2 (right), why is self-consistency particularly important for larger models, given the clear difference between PQM and PQM+SC only in the 70B model? Providing additional insights here would be very useful.
- I would recommend adding some bold numbers to Table 3 also (minor issue)

**Questions:**

- The best results on MATH 500 are achieved with MetaMath-Mistral-7B when using zeta = 4, while for LLaMA-3-70B-instruct, zeta = 2 yields the highest performance. Interestingly, this relationship is reversed for the GSM-Plus benchmark. Is there some intuition behind why the relative importance of intermediate steps has such a model-dependent influence? This trend is also visible in Table 3.
- Similarly, is the zeta value of 2 required for larger models in general, or is this just a quirk of llama-3?
- Intuitively, PRMs interpret all subsequent steps after the first mistake as wrong. How does the proposed method handle these situations?

---

> ### Author Response · Authors · 2024-11-16
> **Response to Reviewer QeSG**
>
> We thank the reviewer for the thorough comments and suggestions. We are deeply encouraged the reviewer recognized the strengths of our work from various perspectives. Below we respond to your comments and questions in detail.
>
>
> > Presentation suggestions.
>
> Thank you for your detailed suggestions. We agree that adding more intuition could enhance Section 3. To address this, we will incorporate higher-level guidance at the start of the section, providing readers with a clearer conceptual overview before diving into the technical details. In addition, we will bold/underline the best/second-best number in Table 3.
>
> > Why self-consistency is particularly effective for 70B model?
>
> That's a great observation! Self-consistency (SC) works by sampling multiple trajectories and selecting the final answer that appears most frequently. **Its performance is therefore closely tied to the capabilities of the underlying sampling policy**. When sampling solutions with a highly capable model, such as Llama3-70B in our experiments, the probability of selecting correct answers across multiple samples by SC is significantly increased. Therefore, the large capacity model tends to reinforce the effectiveness of SC, leading to the increased performance gap observed in the figure. As suggested, we will add this explanation in our revised draft.
>
> > Is there some intuition behind why the relative importance of intermediate steps has such a model-dependent influence?
> Is the zeta value of 2 required for larger models in general, or is this just a quirk of llama-3?
>
>
> Thank you for the insightful question. To better understand this, we further conduct an evaluation on Eurus-8\*22B-NCA, another big-scale strong LLM on reasoning, to see whether this trend holds consistently for larger models. As shown in the table, PQM with $\zeta=8$ achieves the best performance. Hence from current experimental results, there is no direct relationship between the policy model's scales and the optimal $\zeta$ value. Nevertheless, PQMs with moderate $\zeta$ values generally achieve higher performance than prior methods. In practice, we do recommend that practitioners validate this margin hyperparameter to achieve optimal performance in their specific settings.
>
> | PQM| @8 | @16| @32| @64| @128|
> | :-----: | :-----: | :-----: |:-----: |:-----: |:-----: |
> | $\zeta=2$| 45.8 |47.6 |45.0|46.4|45.4|
> | $\zeta=4$| 48.8 | 47.8|48.2|52.4|50.4|
> | $\zeta=8$|  51.4|49.2 |49.4|53.6|51.2|
>
> > PRMs interpret all subsequent steps after the first mistake as wrong. How does the proposed method handle these situations?
>
> Theoretically, in our definition and derivations, if the Q-value of step $a_i$ is low, it indeed means that subsequent steps have a large possibility of being wrong. However, in our theoretical framework, we do not impose a strict requirement that all steps following the first mistake are incorrect. Since the policy model has the potential to correct prior errors in subsequent steps, we only assume in Assumption 3.1 that $P(\tau|s)>P(\tau|\overline{s})$, which means achieving the correct answer from a correct state is much easier than from a wrong state.
>
> Practically, in the current automated data construction process, steps following the first mistake are uniformly labeled as incorrect. Our research focuses on modeling rather than improving data quality. However, in Lines 294-303, we have discussed this limitation of the training corpus and adapted our theoretical loss function to a practical version to fit this situation. We believe that a more reliable training corpus can further enhance the performance of our PQM.

---

> ### Comment · Reviewer_QeSG · 2024-11-20
>
> Thank you for taking the time to respond to my review and providing further clarifications.
>
> I appreciate your willingness to provide some additional intuition. Adding some discussion on self-consistency, particularly for large models, would be great as, given the additional experiments, there seems to be a trend there that could be interesting.
>
> Overall, the author's response has largely addressed my questions; however, I will maintain my score.

---

> > ### Author Response · Authors · 2024-11-21
> >
> > We greatly appreciate your comments and suggestions. We will supplement more relevant discussions about self-consistency according to your suggestion.
> >
> > Thank you once again for your thorough review and constructive feedback.

---

### Official Review · Reviewer_oPLj · 2024-11-04

**Soundness:** 3
**Presentation:** 3
**Contribution:** 3
**Rating:** 8
**Confidence:** 2

**Summary:**

The paper proposes Process Q-Value model (PQM) a framework which uses Q-value instead of immediate rewards in the Process Reward Modeling (PRM).

Unlike PRM, the proposed framework allows to model inter-dependencies among reasoning states. Moreover, the authors show that the existing PRM framework can be cast as a special case of PQM.

The authors provide an extensive empirical study of the proposed method and demonstrate the clear benefit of the proposed method.

**Strengths:**

Overall, the paper is overall well written (see one remark in Weaknesses) and easy to follow.

The paper extends the existing PRM framework to a more general PQM framework which uses Q-values instead of intermediate rewards, which allows to capture the dependency between reasoning states (rather than having these states to be independent). This extension is natural and well motivated.

The paper provides empirical study highlighting the effectiveness of the proposed method. Moreover, the paper provides ablations for the introduced hyperparameters. For example, Table 3 suggests U-shape behavior of the $\zeta$ parameter with good values being in the middle. Overall, it seems that there is a sufficiently large range for this parameter which leads to good performance.

**Weaknesses:**

A suggestion to slightly improve presentation. In Section 3.3, it would be helpful to outline the overall objective for the theorem 3.5 (what do we want to prove any why), and then outline the plan for this proof (why we need other lemmas).

From the results, it is unclear why (Figure 2) the gap in performance between SC+PQM and PQM is larger as we go to the right. It would be helpful if the authors add explanations of why they believe it is happening.

**Questions:**

Why is the gap in performance between PQM and SC+PQM increases as we move to the right in Figure 2?

---

> ### Author Response · Authors · 2024-11-16
> **Response to Reviewer oPLj**
>
> We thank the reviewer for the constructive comments and suggestions. We are deeply encouraged the reviewer recognized the strengths of our work from various perspectives. Below we respond to your comments and questions in detail.
>
> > Presentation Suggestion
>
> Thank you for your valuable feedback. We will revise our framework presentation accordingly. To improve clarity, we will introduce our training objective at the beginning of Section 3.3, then outline the step-by-step approach to our proof, followed by a detailed presentation of the proofs.
>
> > Why does the gap in performance between PQM and SC+PQM increase as we move to the right in Figure 2?
>
> That's a great observation! Self-consistency works by sampling multiple trajectories and selecting the final answer that appears most frequently. **Its performance is therefore closely tied to the capabilities of the underlying sampling policy**. When sampling solutions with a highly capable model, such as Llama3-70B in the right of Figure 2, the probability of selecting correct answers across multiple samples by SC is significantly increased. Therefore, the large capacity model tends to reinforce the effectiveness of SC, leading to the increased performance gap observed in the figure. We will add this explanation in our revised draft.

---

> > ### Comment · Reviewer_oPLj · 2024-12-02
> > **Acknowledgement**
> >
> > I would like to thank the authors for their response and for clarifying my question about the performance gap between PQM and SC+PQM. I am keeping my current score.

---

> > > ### Author Response · Authors · 2024-12-02
> > >
> > > Thank you so much for your appreciation of our work. We are delighted that our responses can clarify your questions. We will definitely implement your valuable suggestions into our final manuscript.

---

### Official Review · Reviewer_SdVZ · 2024-11-12

**Soundness:** 2
**Presentation:** 2
**Contribution:** 3
**Rating:** 5
**Confidence:** 4

**Summary:**

The paper introduces an algorithm to train a process verifier using a Q-value ranking objective. In particular, they split the intermediate steps in a generated trajectory into steps with high and low Q-values and optimize a ranking loss with margins. The authors empirically show that the learned PQM is better for best-of-N, compared to prior works like Wang et. al., Lightman et al., that use a classification based loss to train the Q-function.

**Strengths:**

- The proposed ranking loss for training PRMs empirically improves best-of-N on MATH500 dataset, for multiple base LLMs. In particular, it outperforms prior work Wang et. al., that trains the PRM with a BCE loss.
- The analysis in Section 4.3 is insightful and shows that using a margin based ranking (ablating on $\zeta$) improves best-of-N performance when using the PQMs as an ORM (taking the min score over individual steps).

**Weaknesses:**

- The trained verifier is only used for best-of-N which is not its most promising use case. Evaluating its efficacy for beam-search where the PQM ranks intermediate generations is needed to demonstrate why practitioners should train PQMs.
- Training with the binary cross-entropy loss, where the labels for each prefix are the expected future reward (some value between 0 and 1), will also distinguish prefixes across problems, and maybe the difference in expected rewards for prefixes can be accentuated with an appropriate scaling factor on the expected reward. In theory, it is unclear why the model trained with ranking loss should lead to a more generalizable solution that the one trained with this version of the classification loss. Essentially, it is unclear why the ranking loss should lead to a solution different from the one trained by Wang et al., when the model is trained on the population objective. I believe that the empirical results suggest otherwise, but this observation lacks formal or intuitive justification.
- It is unclear if BCE and $Q_\sigma$ models induce very different rankings on the Q-values of the steps. From the qualitative example in Table 4, it seems that they do not differ by a lot. An empirical study is needed to understand if they result in very different rankings on the steps in test examples. And if the rankings do not differ by a lot, then some post-hoc calibration of the BCE solution should also perform as well as PQM. And if they do differ in rankings, then it is more interesting, and important to study why this happens.
- Are PQMs more sample efficient than BCE? Some analysis on training PQMs with different dataset sizes would help clarify. Also, what is the best data collection strategy to train PQMs? Is it just IID samples from the base LLM? Should we sample more solutions per question, or focus on improving coverage over both questions and solutions. Basically, if we have a fixed sampling budget, how should we collect data to train PQMs?
- In a number of places, the objective optimized or algorithm used is unclear (see Questions).

**Questions:**

- What is the precise definition of a correct or incorrect step? Is it based on the Q-value?
- Equation 3 and 4 are unclear, and in general the notation $\bar{a_{1:t}}$ is confusing. For example, since $\tau$ in L222 is composed of $x, a_1, \ldots, a_H$, shouldn't $P(\tau | \bar{a_{1:m}})$ be $0$ in Equation 3?
- In the objective in Eq. 9, how is $Q_{w_t}$ computed, i.e., what is the input to the network? Is it only the question and action, or the whole prefix up until the action $w_t$?

---

> ### Author Response · Authors · 2024-11-16
> **Response to Reviewer SdVZ (Part I)**
>
> We thank the reviewer for the thorough comments and suggestions. Below we respond to your comments in detail.
>
> > Evaluating PRM efficacy for beam-search where the PQM ranks intermediate generations is needed to demonstrate why practitioners should train PQMs.
>
> Thank you for this insightful suggestion. In our current experiments, we focused on the best-of-N setting to align with standard evaluation practices [1][2]. We agree that exploring PQM's effectiveness in beam search scenarios could provide additional value. To further validate the effectiveness of our PQM, we have conducted additional experiments on PQM-guided beam-search as suggested. We set the beam size as 8, the temperature as 0.7. The evaluation is conducted on MATH500 across Llama-3-8B-Instruct and Eurus-7b-sft. The results are reported in the table below, which demonstrate that PQM can more effectively guide the LLM to reason.
>
> | policy model | pass@1 |classification-based PRM-guided| PQM-guided|
> | :-----: | :-----: | :-----: |:-----: |
> | Llama-3-8B-Instruct| 17.2 | 26.4 | **31.6**|
> | Eurus-7b-sft|  19.4 | 24.2 |**29.2**|
>
> [1] Peiyi Wang, Lei Li, Zhihong Shao, R. X. Xu, Damai Dai, Yifei Li, Deli Chen, Y. Wu, and Zhifang Sui. Math-shepherd: Verify and reinforce llms step-by-step without human annotations. arXiv preprint arXiv:2312.08935, 2023a
>
> [2] Liangchen Luo, Yinxiao Liu, Rosanne Liu, Samrat Phatale, Harsh Lara, Yunxuan Li, Lei Shu, Yun Zhu, Lei Meng, Jiao Sun, and Abhinav Rastogi. Improve mathematical reasoning in language models by automated process supervision. arXiv preprint arXiv:2406.06592, 2024.
>
> > Training with the binary cross-entropy loss can also distinguish prefixes across problems, it is unclear why the model trained with ranking loss should lead to a more generalizable solution.
>
>
> Thank you for this thoughtful question. While training with binary cross-entropy loss based on expected future rewards can help distinguish prefixes across problems, our ranking-based objective is specifically designed to capture the relational dynamics of intermediate reasoning states within a trajectory.
>
> Our theoretical framework (detailed in Section 3.4) suggests that a ranking loss, by focusing on the ordinal relationships between reasoning states, aligns more closely with the structure of sequential decision-making tasks. Moreover, **as discussed in Section 3.5, we provide a theoretical foundation showing that classification-based PRM can indeed be cast as a special case of our framework**. This demonstrates that our approach has the generality to encompass classification-based methods while adding flexibility through a ranking perspective. Our empirical results further support this, demonstrating that the ranking-based approach yields significantly stronger performance across benchmarks compared to BCE loss.

---

> ### Author Response · Authors · 2024-11-16
> **Response to Reviewer SdVZ (Part II)**
>
> > It is unclear if BCE and PQM induce very different rankings on the Q-values of the steps according to qualitative examples in Table 4. If they do differ in rankings, why does this happen.
>
> We would like to clarify that models trained with BCE vs PQM indeed produce very different ranking behaviors. We will explain this using both qualitative example and statistical analysis.
>
> We first highlight these behavioral differences qualitatively based on Table 4:
>
> - BCE produces probabilities that are monotonically decreasing for correct steps (step 1: 0.916-> step 2: 0.882 -> step 0.848). This behavior contradicts the desired property established in Theorem 3.5, which proves that values should increase (rather than decrease) for correct reasoning steps.
> - BCE does not produce a large transition in values between correct and incorrect steps. For example, in Table 4, the probability only slightly decreases from 0.848 (step 3) to 0.628 (step 4), failing to sharply differentiate between correct and incorrect steps. In contrast, our PQM framework produces Q-values with a significant drop from correct to incorrect steps, better aligning with the desired behavior. For example, in Table 4, the $Q_\sigma$ value drops substantially from 0.482 to 0.004 between steps 3 and 4.
>
> Statistically, as suggested, we conducted an empirical study to confirm whether BCE and PQM result in different rankings on test steps. We calculated the proportion of solutions where classification-based PRM and PQM produce the same rankings across steps. **Only 29.18% of solutions shared the same rankings**, indicating a significant behavioral difference between BCE and PQM. Furthermore, when comparing rankings across different solutions for the same question (Best-of-N results), we observed that **0% of test questions had identical rankings among steps of 128 solutions**, confirming that PQM’s ranking-based approach induces unique ordering behavior compared to BCE.
>
> The primary reason for these differences lies in the nature of the objective functions. BCE operates on individual reasoning states independently, treating each state as a binary outcome rather than considering its relation to the state sequence of the solution. As a result, BCE lacks the capacity to explicitly enforce dependencies between reasoning states, leading to less distinct value transition across correct and incorrect states. On the other hand, PQM’s ranking-based loss is designed to optimize the relative quality of each reasoning state in the context of the entire solution, aligning with the optimal ranking proved by our Theorem. PQM thereby reflects an ordinal relationship between reasoning states, providing a clearer separation that allows for more interpretable rankings across steps.
>
>
> > Are PQMs more sample efficient than BCE? Some analysis on training PQMs with different dataset sizes would help clarify.
>
> That is an interesting question! As suggested, we report below the performance for both BCE and PQM, across different dataset sizes. Specifically, we randomly sample 25%, 50%, 75% of the original dataset to train PRMs with $\zeta=4$, and evaluate them on MATH500 sampled by Llama-3-70B-Instruct and MetaMath-Mistral-7B. The numbers in each cell correspond to BON@8/16/32/64/128. The results suggest that **PQM generally outperforms BCE on all ranges of data sizes, and is more sample efficient**.
>
> | data size | BCE(Llama3) | PQM(Llama3)| BCE(MetaMath)|PQM(MetaMath)|
> | :-----: | :-----: | :-----: | :-----: | :-----: |
> | 25%| 37.2/35.6/34.2/34.6/30.0 | 37.4/36.6/37.2/38.4/35.6 |19.6/21.0/18.2/19.0/17.8 |21.4/21.6/19.8/19.8/19.2|
> | 50% | 37.6/35.4/32.6/31.8/29.0 |37.4/36.4/34.4/34.2/32.6| 23.6/24.2/22.8/22.4/19.8 |21.0/22.0/20.2/20.2/19.4|
> | 75% | 40.6/38.8/37.0/38.4/38.8 | 46.8/47.8/47.0/47.2/46.0 |32.4/31.8/34.0/34.6/33.6 |33.4/36.4/37.0/39.6/38.0|
> | 100% |43.6/41.4/41.6/42.4/39.8| 47.2/48.2/50.0/46.0/47.8 |33.6/37.0/39.2/40.8/42.0| 36.2/38.2/41.0/44.2/44.6|
>
>
> > What is the best data collection strategy to train PQMs?
>
> Thank you for the question. While data collection strategy is indeed crucial for effectively training PQMs, it falls outside the primary scope and focus of our paper, which centers on the learning mechanism itself. As mentioned in Lines 857-859, we believe that process reward models could benefit from a more advanced data collection approach. We hypothesize a superior data collection strategy should combine diverse solutions for each question with broad question coverage to maximize generalizability. Exploring optimal data collection strategies for PQMs is a valuable direction for future work.

---

> ### Author Response · Authors · 2024-11-16
> **Response to Reviewer SdVZ (Part III)**
>
> > What is the precise definition of a correct or incorrect step?
>
> We describe this in L39-L44. Specifically, for a trajectory {$x, a_1, a_2,...,a_H$}, where $x,a,H$ represent a question, a reasoning step, and the trajectory horizon. Each reasoning step $a_i$ is generated based on the current reasoning state $s_i=(x, a_{1:i-1})$. If step $a_{i}$ is logically correct considering previous steps and the question $x$, it is a correct step; otherwise, it is a wrong step. We also provide an illustrative example in **Figure 1**, where the first steps are correct and the last three steps are incorrect.
>
> > The notation $\overline{a_{1:n}}$ is confusing. Shouldn't $P(\tau|\overline{a_{1:m}})$ be 0 in Equation 3?
>
> In **line 210-215**, we have shown several cases to exemplify the meaning of this notation, where we use the original notations to represent the correctness of reasoning states and an **overline to indicate the incorrectness of a reasoning state**. For example, $P(\tau|\overline{a_{1:m}})$ in the above question means the possibility to generate a correct trajectory $\tau$ from a _incorrect reasoning state_ $a_{1:m}$.
>
> The value of $P(\tau|\overline{a_{1:m}})$ should not be $0$, since the policy model has a small possibility in subsequent generations to revise the error in $a_{1:m}$ and finally achieve the correct solution. In our theoretical derivations, we do not constrain $P(\tau|\overline{s})=0$, while we only assume mildly $P(\tau|s)>P(\tau|\overline{s})$ in Assumption 3.1, i.e., achieving a correct answer from a correct state is much easier than from a wrong state. This assumption has also been empirically verified in L469-L482. As shown in Figure 3 (right), the empirical probability of $P(\tau|\overline{s})$ is small, but not strictly 0.
>
> > In the objective in Eq. 9, how is $Q_{w^t}$ computed, i.e., what is the input to the network?
>
> As shown in Figure 1, we use the basic RM model structure (an LLM backbone and a value head), the input of $Q_i$ is $(x\oplus a_{1:i-1},a_i)$, i.e., $x\oplus a_{1:i}$. For a trajectory $\tau=\{a_1,\dots,a_H\}$, it only needs a single forward pass to get $Q_1,\dots,Q_H$.

---

> > ### Author Response · Authors · 2024-11-22
> > **follow up**
> >
> > Dear Reviewers SdVZ:
> >
> > We wanted to touch base with you as the deadline for the author-reviewer discussion phase is approaching on November 26. We trust you've had the opportunity to review our rebuttal, and we would be more than happy to address any further concerns you have.
> >
> > These days, we have also updated our draft to incorporate your suggestions. Your feedback has been instrumental in improving our work - thank you again!
> >
> > Best,
> >
> > Authors

---

> > > ### Comment · Reviewer_SdVZ · 2024-11-25
> > >
> > > Thank you for your detailed response to my questions in the review and apologies for the delay in the response on my end. I still have some other questions and concerns that I outline below.
> > >
> > > - The experiments with beam search definitely strengthen the paper, and I would encourage adding more ablations on PRM guided search to the final version of the paper.
> > > - Thank you for clarifying that the PRM trained with BCE loss offers a different ranking than the one trained with ranking loss. Is this also true when you optimize either objective on population data (or alternatively, is this true on the training data)? Or only true on the test set, only when optimizing a finite training setting? I ask this question because a perfectly calibrated model (or Bayes optimal predictor) is realized by optimizing the BCE loss on the population data.
> > > - Almost all of the theoretical analysis (e.g., Theorem 3.5) relies on 1) Assumption 3.1: probability of generating a correct solution trace is higher, when sampling from a logically correct prefix, than from an incorrect prefix 2) If the final answer is correct, then the solution must be logically correct. Have the authors done any empirical evaluation of Assumption 3.1?
> > > - Is the BCE solution worse than the solution of the ranking based objective mainly due to poor calibration? Is some post-hoc calibration of the BCE solution equally good? For example, if we threshold the BCE solution's values on 0.7 in Table 4, then the behavior should be similar to $Q_\sigma$?

---

> ### Author Response · Authors · 2024-11-26
>
> Thanks for taking the time to read our response. We are happy to clarify your questions further. According to your suggestions, we have updated our manuscript.
>
> > More ablations on PRM guided search
>
> Thank you for this valuable suggestion. In addition to the comparison between classification-based PRMs and our PQMs on beam search, we now present further beam search results for PQMs trained with $\mathcal{L}_\textrm{theorem}$ and different $\zeta$ values on MATH500, using the Eurus-7b-sft policy model. These results align with the findings from the Best-of-N experiments, showing that a sufficiently large range of $\zeta$ leads to strong performance in PRM-guided beam search, with optimal values typically falling in the middle of the range.
>
> | Objective | $\zeta=1$ | $\zeta=2$ | $\zeta=4$ | $\zeta=8$ | $\zeta=16$ |
> | :-----: |:-----: | :-----: | :-----: |:-----: |:-----: |
> | $\mathcal{L}$|26.4 | 27.8 | 28.8 | 28.4 | 25.6|
> | $\mathcal{L}_\textrm{theorem}$| 24.8 | 26.0 | 28.0 | 28.2 | 26.6|
>
> We have supplemented this ablation experiment on PRM-guided beam search in the newest version (Table 8).
>
>
> > Is different ranking behaviour hold true on the training data?
>
> As suggested, we compare the ranking behaviours between classification-based PRMs and our PQMs on training data. We randomly sample 2048 cases from the training set for this experiment. Statistically, classification-based PRMs and PQM yield different rannking behaviours on $62.79$% training cases. The primary reason for these differences lies in the nature of the objective functions, where PQM’s ranking-based loss is designed to optimize the relative quality of each reasoning state in the context of the entire solution, but BCE lacks the capacity to explicitly enforce dependencies between reasoning states.
>
> We appreciate your insightful demonstration that optimizing the BCE loss on the population data with infinite training data can also approximate Q-value. **This has been already encompassed by our theory. In section 3.5, we prove BCE loss is a special case under our theoretical framework, and can also approximate Q-value without bias. (Line 322-323)**
>
> Due to the superiority of PQM that can capture the interdependencies of intermediate reasoning states, our PQMs can achieve higher sample-efficieny and better performance practically.
>
> > Is there any empirical evaluation of Assumption 3.1?
>
> Yes, **we have thoroughly verified Assumption 3.1 in Section 4.3**. We kindly ask you to refer to **Line 473-477** for more details. **Figure 3** demonstrates that when conditioned on a correct reasoning state, there is a higher probability of generating a correct subsequent step or completing a correct trajectory.
>
> > Is the BCE solution worse than the solution of the ranking based objective mainly due to poor calibration? Is some post-hoc calibration of the BCE solution equally good?
>
> As explained in Line 852-866, our ranking-based objective results in a quite different ranking behavior from the classification-based PRM's. Though calibration can adjust the relative magnitude of predicted reward, **calibration preserves the order and cannot change the rankings posthoc** among rewards of different steps.
>
> For example, applying a threshold of 0.7 to Table 4 does not mitigate the incorrect ranking over first three steps according to our theory, where the Q-values of Steps 1 to 3 (all correct steps) should ideally follow an increasing trend ($Q_1^* < Q_2^* < Q_3^*$), but **BCE displays the erroneous ranking, with decreasing values (0.916 > 0.882 > 0.848)**.
>
> Importantly, ranking relationships, rather than absolute values, are critical in tasks like Best-of-N sampling or PRM-guided beam search. Since the ranking behaviors are quite different between our PQMs and classification-based PRMs (Line 859-875), post-hoc calibration on the BCE solution cannot achieve the same performance as our ranking-based objective.

---

> > ### Comment · Reviewer_SdVZ · 2024-11-26
> >
> > Thank you for adding the ablations and for the responding to the other questions. Regarding the ranking on training data, if the BCE solution is trained till the interpolation regime, then the trained PRM should simply predict the Q-value in the training data, right? And the BCE model is also given as input the full context, just like the ranking model, if I understand correctly. In this case there should be no difference in rankings?

---

> ### Author Response · Authors · 2024-11-27
>
> Thank you for the follow-up questions. We would like to clarify some key points and address potential misunderstandings.
>
> First, existing training corpora for PRMs typically provide binary correctness labels for each step (e.g., [1, 1, 1, 0, 0, 0, 0]) rather than specific value labels. **Without our theory, there is no ranking on training data.**
>
>
> Additionally, in practical scenarios with finite training samples, most reasoning states $s_i$ appear only once in the training corpus due to the large action space of LLMs. This causes the observed frequencies to deviate significantly from the expected distribution $\mathbb{E}~\pi(s_i)$. As a result, even when a PRM trained with BCE reaches the interpolation regime, it generally predicts near $1$ or $0$ for steps of training samples. For instance, given a training sample with step labels  [1,1,1,0,0,0,0], training by BCE loss with zero training error would make PRMs' predict values to be [1,1,1,0,0,0,0]. This results in very different ranking behaviors between two methods:
>
> > **BCE's Q-value ranking**: $Q_1=Q_2=Q3>Q4=Q_5=Q_6=Q7$
>
> > **PQM's Q-value ranking**: $Q_1 < Q_2 < Q_3 \gg Q4 > Q_5 > Q_6 > Q_7$
>
> PQM’s ranking is more coherent with the Q-function definition in Eq. (2) of our paper, explicitly capturing the interdependencies among reasoning states.
>
> Moreover, in practice, training may not always reach zero training error due to model capacity, optimization constraints, and noisy training data. **Taking these factors into account, BCE-trained PRMs exhibit significantly different and less consistent ranking behaviors compared to PQMs**. The differences can be seen in our case studies and behavior analyses as in Table 4 (where BCE produces erroneous ranking with $Q_1 > Q_2 > Q_3$) and Line 859-875.
>
> To summarize, we provide an intuitive explaination for why our PQM could perform better. **With our theory, PQM can explicitly optimize the ranking relationship and interdependencies among different states, hence able to more accurately approximate Q-values as defined in Eq.2 with higher sample-efficiency**.

---

> > ### Comment · Reviewer_SdVZ · 2024-11-27
> >
> > Thank you for the clarification, it would be good to make this more clear in the paper too, since I was under the impression that the BCE solution is trained by minimizing the binary cross-entropy loss at each prefix $x$ in the dataset, where the target for the prefix is a scalar $\in [0, 1]$, i.e., expected outcome reward $E_{\pi(y \mid x)} I(x,y)$ on future completions $y$ generated by conditioning on $x$, and this value is computed through Monte-Carlo rollouts. If this is not the case, how is this computed to obtain the rankings on the correct and incorrect steps for the ranking loss?

---

> > > ### Author Response · Authors · 2024-11-28
> > >
> > > Thank you for the follow up questions. We are glad our clarifications helped.
> > >
> > > > Demonstration about BCE loss and binary label.
> > >
> > > We kindly ask you to refer to Section 2 (**Line 126-134**), where we have already introduced the BCE loss and the data format with 0-1 labels. This section provides a detailed explanation of how binary correctness labels are used in existing PRM.
> > >
> > > > How to obtain the rankings on the correct and incorrect steps for the ranking loss?
> > >
> > > We would like to clarify a potential misunderstanding here.
> > > **By virtue of our new Theorem 3.1, we do not need explicit estimation of expected outcome reward $E_{\pi(y \mid x)} I(x,y)$. Instead, we rely solely on binary correctness labels to construct ranking relationships over a trajectory**. This is precisely how our work reframes PRM from a classification-based problem to a ranking-based problem, which constitutes the core novelty and significant contribution of our work.
> > >
> > > Here, we briefly encapsulate our theory, then show some specific examples to illustrate how to construct ranking relationships with only binary labels, and briefly summarize the derivations of our theory.
> > >
> > > ### === Theory and examples ===
> > >
> > > According to Thereom 3.5, for a trajectory {$a_1,a_2,\dots,a_H$}, if $C=${$c_i$} is the index set of correct steps and $W=${$w_i$} is the index set of wrong steps (i.e. $a_{c_i}$ is a correct step with label $1$, $a_{w_i}$ is a wrong step with label $0$), we establish a ranking relationship among $Q_i = Q(a_{1:i-1},a_i)$ as follows.
> > >
> > > $Q_{w_{|W|}}< \dots <Q_{w_2}<Q_{w_1}\ll Q_{c_1}<Q_{c_2}<\dots<Q_{c_{|C|}}$
> > >
> > > where $|C|,|W|$ is the length of correct,wrong index list, and $|C|+|W|=H$, i.e. the total step number. Hence, only with the 0-1 correctness label, we could establish a Q-value ranking for a trajectory.
> > >
> > > We provide some specific examples as shown in the table below. The first row means that if a training sample with binary labels [1,1,1,0,0], by our theory, we will **train the predicted Q-value $Q_1,\dots,Q_5$ to satisfy $Q_5<Q_4\ll Q_1<Q_2<Q_3$ by a ranking-based loss.**
> > >
> > > | binary correctness label| ranking relationship by our theory|
> > > | :-----: |:-----: |
> > > |[1,1,1,0,0]|$Q_5<Q_4\ll Q_1<Q_2<Q_3$|
> > > |[1,0,0,1,1]|$Q_3<Q_2\ll Q_1<Q_4<Q_5$|
> > >
> > > ### === Derivation of Theorem 3.1 ===
> > > To derive Theorem 3.1, we
> > > 1. Introduce Assumption 3.1, which is empirically verified in Section 4.3, and Bayesian factorization as in Eq(3) & Eq(4).
> > > 2. Establish pairwise relationships between earlier and later steps:
> > >      - Lemma 3.3: An earlier correct step has a smaller Q-value than a later correct step, and an earlier wrong step has a larger Q-value than a later wrong step.
> > >      - Lemma 3.4: Analyze the relationship between the first correct and the first wrong step.
> > > 3. Conclude an integral ranking relationship over the entire trajectory, as formally stated in Theorem 3.5.
> > >
> > >
> > > ### === Advantages of our framework ===
> > > Our framework bypasses the need for explicit estimation of $E_{\pi(y \mid x)} I(x,y)$, which is very computationally expensive through Monte-Carlo rollouts, and instead construct a ranking relationship for every training data with only binary correctness labels. Compared to BCE loss under the same data with binary labels, PQM is able to
> > > - Capture interdependences among different reasoning states within a single trajectory.
> > > - Achieve higher practical performance and sample efficiency.

---

> > > ### Author Response · Authors · 2024-12-01
> > >
> > > Dear Reviewer SdVZ,
> > >
> > > As the discussion period ends soon on December 2, we wanted to kindly remind you to share any final questions or comments. We hope our response has helped clarify the contributions of our work, and we'd be happy to answer any remainder questions. Thanks again for your time and for actively engaging in this dialogue.
> > >
> > > Best,
> > >
> > > Authors

---

> > > > ### Comment · Reviewer_SdVZ · 2024-12-01
> > > >
> > > > Thank you for the clarification on the labels being used. This was not clear to me from Section 3 and I would encourage adding the above example to the paper. So, how are we getting the binary labels for the steps. My understanding was that this was identified by computing $E[I(x, y)]$ before and after the step and then setting the steps with drops in $E[I(x, y)]$ as incorrect (-ve advantage) and increase in $E[I(x, y)]$ as correct (+ve advantage)?

---

> ### Author Response · Authors · 2024-12-01
>
> We fully agree with your suggestion regarding the example and will incorporate it into our paper for greater clarity. The binary label generation process is detailed in **Appendix A (Lines 704–723)**, proposed by Wang et al. [1].  Specifically, for each step in a trajectory, multiple completions are sampled. If any completion leads to the correct final answer, the step is labeled as 1; otherwise, it is labeled as 0.
>
> To ensure fair comparisons, both BCE and PQM are trained using the same binary labels from the original Math-Shepherd dataset. These 0-1 labels do not involve value estimation, and simply indicate the binary correctness of each individual step.
>
> We believe that process reward models could benefit from a more advanced data collection approach, but we would like to kindly remind you that data collection strategy falls outside the primary scope and focus of our paper, which centers on the learning mechanism itself.
>
>
> [1] Peiyi Wang, Lei Li, Zhihong Shao, R. X. Xu, Damai Dai, Yifei Li, Deli Chen, Y. Wu, and Zhifang Sui. Math-shepherd: Verify and reinforce llms step-by-step without human annotations. arXiv preprint arXiv:2312.08935, 2023

---

> ### Comment · Reviewer_SdVZ · 2024-12-02
>
> Thank you so much for all the clarification and responding patiently to all my queries!
>
> My final evaluation is as follows. This work shows that the ranking loss is able to more clearly distinguish incorrect steps from correct steps, where the incorrect step and correct steps are defined using binary labels (step is incorrect when no rollout from multiple conditional rollouts leads to the correct answer). While the results look promising, I have two concerns.
>
> - First, I feel that the comparison with BCE (which is a much simpler training objective) is not very fair. For example, if we simply computed the Monte Carlo estimate under the rollout policy at each step and use that to train PRM with BCE loss (here the label is no longer binary), it should be able to model the dependencies between different steps in the same trajectory.
>
> - Second, their theoretical model assumes (Assumption 3.1) something about the distribution of the optimal policy. It is unclear what this optimal policy -- I would think it is the one that always generates the correct answer from any state, but that does not seem to be the case in their model. Also, why should the PQM learn to predict the Q-values of this optimal policy? Since, we typically use PQMs for test-time beam search or as dense reward models in online RL, where the roll-in policy is not optimal.
>
> Even though there are a couple of technical gaps (like the ones discussed above), I appreciate the empirical efforts to add in new results with beam search, and ablations on different values of the margin $\zeta$. In light of this, I will raise my score to 5. Even though 5 is leaning towards reject, I would not be *strongly* opposed to the acceptance of this work!
>
> Edit: I will also consider raising the score further post discussion with other reviewers and AC.

---

> ### Author Response · Authors · 2024-12-03
>
> Thank you very much for reviewing our responses and for raising the score. We appreciate your engagement and would like to respond to the remaining concerns.
>
> --------
> > Clarification on baseline
>
> First, we believe **our comparison with BCE loss is fair because it _strictly_ adheres to the established baseline methodology and implementations**, following OpenAI’s work [1] and the dataset protocol employed in [2]. By aligning with these well-established practices, our comparisons are consistent with community standards and comparable to existing publications.
>
> **Moreover, your suggestion to use Monte Carlo estimates as scalar soft labels has been already discussed and compared in our paper, denoted as $MSE_{MCTS}$ in our Table 1**. The implementation and MC sampling strategy follows prior work [3]. Furthermore, as shown in Section 5.2 of Math-Shepherd [2] (the paper describing our training data), soft-label and hard-label approaches perform similarly. The reasons why soft labels lead to similar or even worse performance are **discussed in Lines 400–404** of our paper. We summarize the reasons as follows: (1) MCTS introduces significant computational costs, and the large action space of LLMs makes sufficient exploration challenging. (2) The sampling policy often deviates significantly from an optimal policy, resulting in biased estimates for soft labels.
>
> Since 0-1 label with BCE loss is the majorty of previous works, and results in a better performance pratically, we highlight the comparison between our approach and this setting.
>
> **To summarize, we have demonstrated that PQM outperforms traditional PRM trained by 0-1 labels and soft labels** using the same amount of training data. By transforming _binary_ classification-based BCE loss to a _continuous_ ranking-based loss, we bypass the overheads of MCTS search, but allow PRM to capture the interdependencies among reasoning states, resulting in a higher performance and sample-efficiency.
>
> In the table below, we extract a part of experimental results **from Tabel 1** in our paper to show the results of PQM and $MSE^{MCTS}$ which trains PRM with soft-label. The numbers in each cell represent BON@8/16/32/64/128.
>
> | Methods| MATH(metamath) |MATH(Llama)| GSM-Plus(metamath)| GSM-Plus(Llama)|
> |:---: |:---: |:---: |:---: |:---: |
> |PQM | 36.2/38.2/41.0/44.2/44.6|47.2/48.2/50.0/46.0/47.8 |62.04/63.58/64.50/64.96/65.20 | 72.54/73.25/73.38/72.79/71.96
> |PRM trained with soft label| 24.2/25.2/26.4/25.0/27.0|36.2/38.2/41.0/44.2/44.6 | 50.91/51.67/50.08/49.58/49.79|62.04/63.58/64.50/64.96/65.20|
>
>
> [1] Wang P, Li L, Shao Z, et al. Math-shepherd: A label-free step-by-step verifier for llms in mathematical reasoning. arXiv preprint arXiv:2312.08935, 2023.
>
> [2] Lightman H, Kosaraju V, Burda Y, et al. Let's verify step by step. arXiv preprint arXiv:2305.20050, 2023.
>
> [3] Zhang D, Zhoubian S, Hu Z, et al. Rest-mcts*: Llm self-training via process reward guided tree search. arXiv preprint arXiv:2406.03816, 2024.
>
> ------
> > It is unclear what this optimal policy in Assumption 3.1. Is it the one that always generates the correct answer from any state. Why should the PQM learn to predict the Q-values of this optimal policy?
>
> **Definition of the optimal policy**. To clarify, optimal policy in our framework is any final model trained by _suitable_ RL algorithms. Hence, a policy that always generates the correct answer from any state is only a special case, which is an idealized and often unattainable one. The optimal policy in our framework reflects the distributional property that sampling from a logically correct prefix is more likely to yield correct completions than sampling from an incorrect prefix (**Lines 202-204**). This relaxed assumption makes our model more practical and broadly applicable across different problem settings. As we have demonstrated empirically in Section 4.3, our Assumption 3.1 can be satisfied in real world.
>
> For the reasons why PQM should learn to Q-values of the optimal model, **as demonstrated in Lemma 3.2, only the Q-value of the optimal policy can function the same as an ideal reward function.** This is a key theoretical insight: the Q-values provide a consistent and scalable way to approximate the underlying reward dynamics, enabling PQM to effectively model interdependencies among reasoning states.

---

### Author Response · Authors · 2024-11-16
**General Response**

We thank all the reviewers for their time and valuable comments. We are encouraged to see that reviewers find our PQM framework to be **interesting**, **well-motivated**, making a **strong and compelling contribution** to process reward modeling (oPLj, QeSG, qfhZ). Reviewers appreciated our **comprehensive experiments** and **insightful** analyses (SdVZ, oPLj, QeSG, qfhZ, 1S6Q). Additionally, several reviewers noted that our paper is **clear and well-written** (oPLj, QeSG, qfhZ), with a structured presentation of both theoretical and empirical results.


As recognized by multiple reviewers, the significance of our work can be summarized as follows:

- Our PQM framework introduces a novel approach to process reward modeling by optimizing Q-value rankings, which enhances the handling of interdependencies among reasoning states---a key advancement over traditional methods. This theoretical framework is natural and well-motivated. Notably, we cast prior classification-based PRM as a special case under our theoretical framework.

-  Our extensive empirical evaluations across various sampling policies, language model backbones, and reasoning benchmarks show that PQM outperforms classification-based PRMs.

- We have conducted comprehensive ablation studies and provided detailed analyses on different loss designs and hyperparameters, further confirming PQM’s practical efficacy and theoretical advantages.

In addition, we greatly appreciate the constructive feedback from the reviewers, which further strengthens our work. Altogether, these contributions position PQM as an impactful solution for advancing process reward modeling in complex reasoning tasks, with promising implications for future research.

Below, we address each reviewer’s comments point by point.

---

> ### Author Response · Authors · 2024-11-22
> **revision summary**
>
> **Summary of revision**
>
> Based on the reviewers' feedback, we have revised our manuscript, with major changes highlighted in color. Below, we summarize the key revisions:
>
> [R1] Added two additional experiments in the appendix: (1) a data efficiency experiment comparing BCE and PQM across different data sizes, and (2) PQM-guided beam search.
>
> [R1] Expanded the appendix with a detailed discussion comparing ranking behaviors between PRMs with BCE loss and our PQM approach.
>
> [R2, R3] Revised Section 3.3 to adopt a clearer top-down structure, beginning with the overall objective and proof outline.
>
> [R3] Highlighted the best results in Table 3 for clarity.
>
> [R2, R3] Added clarification of the performance gap between PQM and PQM+SC (Figure 2).
>
> We believe these revisions have further strengthened our manuscript and addressed the key concerns raised. We thank reviewers again for the helpful feedback!

---

### Meta-Review · Area_Chair_CAG1 · 2024-12-19

**Metareview:**

This paper presents a new approach to process reward modeling called PQM that focuses on optimizing Q-value rankings rather than treating the problem as a simple classification task. By doing so, PQM better captures the inherent relationships between steps in a reasoning process. This strength is supported by the authors' extensive evaluation across different models and benchmarks, demonstrating PQM's superior performance.

However, the reviewers raised valid concerns about the paper's clarity on certain aspects, such as sample efficiency compared to existing methods, lack of PQM-guided search, and the optimal data collection strategies for PQM. The authors conducted several additional experiments to address these weaknesses. Some weaknesses still remain (as pointed by SdVZ) such as needing more thorough comparisons with soft MC labels, which have been shown to be better than binary labels [1]. Nevertheless, I recommend acceptance as I believe PQM would still be a useful addition to ICLR.

Suggestion: Some concurrent line of work might be worth discussing is work on training implicit PRMs (still used as ORM) using generative verifiers / RMs [2, 3] as well as work on PRMs based on advantages instead of values functions [4].

[1] Improve Mathematical Reasoning in Language Models by Automated Process Supervision. (see Table 4.3)
[2] Generative Verifiers: Reward Modeling as Next-Token Prediction
[3] Critique-out-Loud Reward Models
[4] Rewarding Progress: Scaling Automated Process Verifiers for LLM Reasoning

**Additional Comments On Reviewer Discussion:**

Most reviewers seem to be in favor of acceptance after discussion, except two reviewers. I am ignoring the review from 1S6Q as it seems to contain weaknesses for another paper. The rebuttal period saw a good discussion between the authors and the expert reviewer SdVZ.  SdVZ questioned the theoretical justification for the ranking loss, the differences in ranking behavior compared to classification-based approaches, sample efficiency, and the limited use case of the trained verifier for only Best-of-N. The authors responded by clarifying the motivation behind their design choices, providing further analysis of ranking behaviors, and reporting performance across different dataset sizes. They also expanded their experiments to include PQM-guided beam-search. This responsiveness to feedback, combined with the paper's strengths, makes me lean towards accept.

---

### Decision · Program_Chairs · 2025-01-22

Accept (Poster)